# Surface charge as activity descriptors for electrochemical $CO_2$ reduction to multi-carbon products on organic-functionalised Cu

Carina Yi Jing Lim[1,2,8], Meltem Yilmaz[1,3,8], Juan Manuel Arce-Ramos[4,8], Albertus D. Handoko[1,8] ✉, Wei Jie Teh[5], Yuangang Zheng[1], Zi Hui Jonathan Khoo[1], Ming Lin[1], Mark Isaacs[6], Teck Lip Dexter Tam[7], Yang Bai[1], Chee Koon Ng[1], Boon Siang Yeo[5], Gopinathan Sankar[3], Ivan P. Parkin[3], Kedar Hippalgaonkar[1,2], Michael B. Sullivan[4], Jia Zhang[4] ✉ & Yee-Fun Lim[1,7] ✉

Intensive research in electrochemical $CO_2$ reduction reaction has resulted in the discovery of numerous high-performance catalysts selective to multi-carbon products, with most of these catalysts still being purely transition metal based. Herein, we present high and stable multi-carbon products selectivity of up to 76.6% across a wide potential range of 1 V on histidine-functionalised Cu. In-situ Raman and density functional theory calculations revealed alternative reaction pathways that involve direct interactions between adsorbed histidine and $CO_2$ reduction intermediates at more cathodic potentials. Strikingly, we found that the yield of multi-carbon products is closely correlated to the surface charge on the catalyst surface, quantified by a pulsed voltammetry-based technique which proved reliable even at very cathodic potentials. We ascribe the surface charge to the population density of adsorbed species on the catalyst surface, which may be exploited as a powerful tool to explain $CO_2$ reduction activity and as a proxy for future catalyst discovery, including organic-inorganic hybrids.

Synergised efforts in recycling, energy recovery, biomass utilisation, and $CO_2$ capture and utilisation (CCU) are required to realise a circular carbon economy with net-zero greenhouse gas emissions[1]. Electrochemical $CO_2$ reduction ($CO_2$RR) is arguably one of the more attractive CCU approaches because of its ability to tailor the catalytic reaction towards higher-value chemical products[2], whilst tapping off-peak electricity on demand. However, the feasibility of $CO_2$RR is contingent on improvement in the stability, Faradaic and energetic efficiencies of the process[3].

[1]Institute of Materials Research and Engineering, Agency for Science, Technology and Research (A*STAR), 2 Fusionopolis Way, Innovis, Singapore 138634, Singapore. [2]School of Materials Science and Engineering, Nanyang Technological University, 50 Nanyang Avenue, Singapore 639798, Singapore. [3]Department of Chemistry, University College London, 20 Gordon Street, London WC1H 0AJ, UK. [4]Institute of High Performance Computing, Agency for Science, Technology and Research (A*STAR), 1 Fusionopolis Way, Connexis, Singapore 138632, Singapore. [5]Department of Chemistry, National University of Singapore, 3 Science Drive 3, Singapore 117543, Singapore. [6]Research Complex at Harwell, Rutherford Appleton Laboratory, Harwell Science and Innovation Campus, Didcot, Oxfordshire OX11 0FA, UK. [7]Institute of Sustainability for Chemical, Engineering and Environment, Agency of Science, Technology and Research (A*STAR), 1 Pesek Road, Singapore 627833, Singapore. [8]These authors contributed equally: Carina Yi Jing Lim, Meltem Yilmaz, Juan Manuel Arce-Ramos, Albertus D. Handoko. ✉e-mail: adhandoko@imre.a-star.edu.sg; zhangj@ihpc.a-star.edu.sg; limyf@imre.a-star.edu.sg

Decades of intensive research on Cu-based catalysts[4,5] have revealed intricate and interwoven relationships of intermediate species adsorption[6], defects[7], catalyst states[8,9], and reaction conditions[10,11] that collectively govern the selectivity and activity of CO2RR. However, experimentally observed activities are still frequently lower than theoretical predictions, suggesting the challenge to understand and accurately model macroscopic and dynamic factors that affect electrocatalysis. Attempts to improve the understanding of the dynamic factors in electrocatalysis have prompted the use of various AC/DC voltammetry techniques beyond established techniques like cyclic/linear sweep voltammetry (CV/LSV) and electrochemical impedance spectroscopy (EIS). Recent investigation on oxygen evolution (OER) catalysis revealed the interplay between applied voltage, accumulated surface charge, catalyst oxidation state change, and reaction rate[12]. However, it is unclear if such a relationship applies to CO2RR, especially to $C_{2+}$ products (molecules with ≥2 carbons) where multiple electrochemical and non-electrochemical steps are involved[13]. Further, CO2RR catalysts (typically transition metals) are not commonly expected to display oxidation state cycling. CO2RR activities have also been shown to be less sensitive to electrochemical surface area (ECSA)[14], and disconnected from the cardinal Tafel values[15].

Herein, we exploit $Cu_2O$-derived Cu with organic functionalisation to unveil the possible role of surface charge in CO2RR to $C_{2+}$ products. Serendipitously, organic functional groups also provide an avenue to circumvent scaling relationships observed with transition metal-only catalysts[16], by providing different bonding configurations to CO2RR intermediates[17]. Increased sensitivity between voltage and catalytic activity is also often observed on organic-functionalised catalysts, a phenomenon that points to surface charge accumulation[12,18]. Histidine is selected as the prime organic functionalisation due to its ability to bind strongly to Cu[19] while containing an imidazole group that can bind and activate $CO_2$[17]. $Cu_2O$-derived Cu functionalised with histidine (Cu-Hist) displays significantly higher $C_{2+}$ product selectivity than the unfunctionalized sample across a wide voltage range. Faradaic efficiencies (FE) of up to 76.6%, corresponding to TOF estimate up to $4.2 \times 10^{-1}\,s^{-1}$ for $C_{2+}$ products, were observed at −2.0 V (vs. reversible hydrogen electrode, RHE throughout), and excellent stability over 48 h. Through careful materials characterisation, in-situ Raman spectroscopy, and density functional theory (DFT) calculations, we discover that the enhanced CO2RR to $C_{2+}$ products on functionalised Cu-Hist is linked to direct intermediate interaction with adsorbed histidine. Strikingly, we uncovered a strong correlation between surface charge magnitude and catalytic activity, suggesting that the electrocatalytic activity in reductive catalysis like CO2RR may also be closely linked to surface charge.

## Results and discussions

### Electrochemical CO₂ reduction data

The CO2RR performance of Cu-Hist was evaluated in a customised H-Cell (Fig. 1, see the "Methods" section and SI Sections 3 and 4). The CO2RR product distribution of Cu-Hist is very peculiar, with a significant preference towards $C_{2+}$ products that stretches even to very wide cathodic potential up to −2.2 V (Fig. 2b). This is in stark contrast to plain $Cu_2O$-derived Cu (Cu-0), where $C_{2+}$ products are quickly overtaken by $CH_4$ and $H_2$ at cathodic potentials beyond −1.2 V (Fig. 2a), consistent with literature[20]. The production of $C_{2+}$ products (particularly ethanol and ethylene) on Cu-Hist rose significantly with increasing voltage, reaching over 20× partial current density compared to Cu-0 (Fig. S4.5b, Table S3.1). Stable formation of $C_{2+}$ products was also observed for 48 h (Fig. 2g). As there is no new metal component in Cu-Hist, and the ECSA of reduced samples are similar (SI Section 3.3), we posit that the significant enhancement in CO2RR selectivity may be caused by a new intermediate stabilisation involving strong chemical interactions with histidine alongside possible contributions from surface-charging effects.

To investigate the possible roles of the different functional groups in histidine, three additional similarly synthesised samples with imidazolium-related functionalisation were added: imidazole (Cu-Im), 2-methylimidazole (Cu-2mIm) and imidazolepropionic acid (Cu-ImPA). The selection of these molecules is based on their relation to histidine (see SI Section 1.3) and the samples' characterisation is presented in SI section 2.

Intriguingly, distinct $C_{2+}$ product selectivity and turnover trends were obtained from the four functionalised samples, despite their closely related surface functionalisation molecules. Enhanced $C_{2+}$ product selectivity was also observed on Cu-Im and Cu-ImPA, albeit with lower peak FE (Fig. 2c, d) and current density ($j$) (Fig. S4.5b) than Cu-Hist. On the other hand, Cu-2mIm behaved more similarly to plain Cu-0, with peak $FE_{C_{2+}}$ occurring at much less cathodic potential of −1.2 V (Fig. 2e) alongside relatively flat $j_{C_{2+}}$ (Fig. S4.5b).

Overall, all functionalised catalysts display >1.5× higher total current density ($j_{tot}$) than Cu-0 (Fig. 2f), despite having similar ECSA (Fig. S3.2). Unlike previous studies on organic functionalised catalysts where enhanced CO2RR is obtained through suppressed HER via a significant increase in surface hydrophobicity[21], a slightly enhanced HER was observed on all our functionalised catalysts at less cathodic potential (−1.0 to −1.2 V, Fig. S4.5a). With more cathodic potentials, HER was maintained at a similar level to uncapped Cu-0, while CO2RR activity is boosted more significantly, allowing us to achieve higher current densities. Overall, the $j_{C_{2+}}$ at −1.6 V or more cathodic potentials follow the trend of Cu-Hist ≫ Cu-Im > Cu-ImPA > Cu-2mIm ≥ Cu-0. Both FE and $j$ trends confirm that functionalisation with imidazole-related molecules indeed enhances CO2RR to $C_{2+}$, especially to $C_2H_4$ at more cathodic potentials. Prior literature links enhanced CO2RR activity on imidazole-related surface functionalisation to the unblocked no. 2 carbon position (C-2) in the imidazole ring (Fig. 1), as it can facilitate the capture of an extra proton with δ+ charge under cathodic potential[22]. However, the presence of the imidazole group alone cannot explain the >27% higher $FE_{ethanol}$ observed on Cu-Hist compared to Cu-Im and Cu-ImPA, as all three functionalisation have the same imidazole group with unblocked C-2 position.

Thus, we hypothesise that the enhanced activity of the Cu-Hist sample to $C_{2+}$ products, particularly to ethanol, may be due to histidine's unique structure. A prerequisite for good surface functionalisation is the stability of the catalyst surface throughout the catalysis and reconstruction process[23]. Both Hist and ImPA are predicted to be the most stable among the five functionalisation molecules as both carboxylate oxygens and proxima nitrogen (N−π) can anchor to Cu (Fig. S1.1)[19]. On the other hand, only N−π nitrogen is available for Im and 2mIm, rendering them less stable. The adsorption of $CO_2$ (or CO) can then be accommodated on the remaining nitrogen sites. Histidine is advantageous, with two N sites available (amine-N and the imidazolate tele-nitrogen, N−τ), compared to ImPA and Im with only one N−τ site. We posit that the presence of organic surface functionalization on the Cu surface can enhance the retention of these intermediate species, thereby favouring the formation of $C_{2+}$ products while suppressing $C_1$ products even at very cathodic potentials. To rationalise the CO2RR enhancement on Cu−Hist, we engaged a multi-pronged approach of in-situ Raman spectroscopy, dynamic voltammetry, and theoretical calculation.

### In-situ Raman spectroscopy

First, we conducted preliminary in-situ Raman to study the adsorption behaviour of histidine under CO2RR conditions. For this purpose, a benchmark in-situ Raman experiment was performed on electrodeposited $Cu_2O$ on Cu substrate in the presence of dissolved 0.025 M histidine in $CO_2$-purged 0.1 M $KHCO_3$ electrolyte (Fig. 3a, see also SI Section 5.1). At open circuit potential (OCP), bands belonging to $Cu_2O$ at 519 and 629 $cm^{-1}$ were identified, along with a possible carbonate band near 1073 $cm^{-1}$. A series of weaker bands around 1009, 1155, 1259,

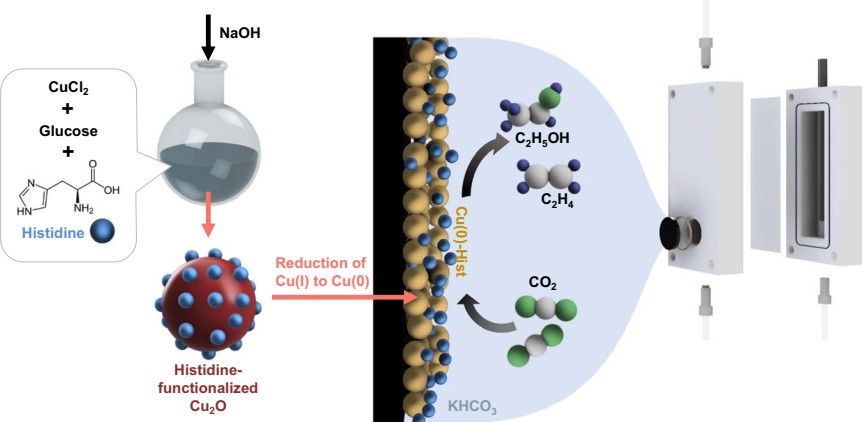

**Fig. 1 | Preparation of histidine functionalised Cu₂O and its subsequent use in CO₂RR.** The histidine remains on the surface after the in-situ reduction of Cu₂O to Cu during catalysis, boosting reaction selectivity towards C₂₊ products. Other organic functionalisations (imidazole, 2-methylimidazole, imidazolepropionic acid, arginine, triazole, and glycine) can be introduced by swapping the reagents during synthesis.

**Fig. 2 | CO₂RR performance comparison between functionalised and organic-functionalised Cu₂O catalysts using different metrics.** FE comparison of Cu₂O-derived Cu with and without surface functionalisation group: **a** Cu-0, **b** Cu-Hist, **c** Cu-ImPA, **d** Cu-Im, **e** Cu-2mIm, and their respective, **f** total current density. **g** Stability of Cu-Hist at −1.6 V over 48 h. Dark blue line represents the total current density ($j_{tot}$). Products were sampled 5 times at $t = 0$, 15, 24, 38, and 48 h for the stability experiment. Periodic dips seen in the $j_{tot}$ plot are due to current re-stabilisation during electrolyte refresh. Error bars represent the standard deviation of three independent measurements.

1485, 1572, and 1640 cm⁻¹ that are consistent with Raman bands of deprotonated L-histidine adsorbed on Cu in the literature (0.1 M NaOH, −0.6 to −1.0 V vs. Ag/AgCl)[24] were also observed. As the cathodic potential is applied, the $CO_3^{2-}$ band at 1073 cm⁻¹ disappeared while the histidine-related bands get significantly stronger. Additional bands at 380, 527, 1110, 1321, 1415, and 2079 cm⁻¹ appeared, which can also be identified as histidine-related bands (SI Section 5.1). The histidine bands persist up to a highly cathodic potential of −1.1 V. Intriguingly, as the potentials are gradually ramped to more cathodic values, the C≡O frustrated rotation and Cu−CO bands (expected around 275

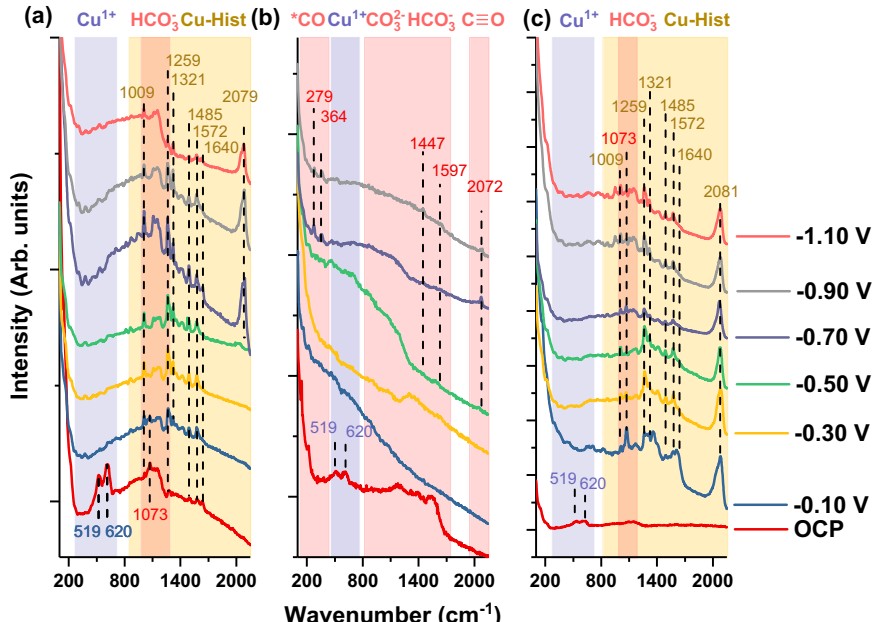

**Fig. 3 | In-situ Raman spectroscopy on bare and histidine functionalised Cu₂O under CO₂RR relevant conditions.** Comparisons were made on three different conditions to ascertain histidine presence during CO₂RR and the expected *CO binding configuration on **a** electrodeposited Cu₂O with added 0.025 M histidine dissolved in the electrolyte. **b** Cu-0 and **c** Cu-Hist. Measurements were stopped at different potentials depending on the vigorousness of the bubbling that disrupts in-situ Raman signal. Electrolyte: CO₂ purged 0.1 M KHCO₃, pH ≈6.7. Red shaded area: expected region of adsorbed CO₂RR intermediate bands. Blue shaded area: expected region of Cu¹⁺ bands. Yellow shaded area: expected region of Cu-Histidine complex bands. Dashed lines are a guide to the eye. Raman bands marked at 1009, 1259, 1321, 1485, 1572, and 1640 cm⁻¹ can be matched with Raman bands of deprotonated ʟ-histidine adsorbed on Cu in alkaline conditions under applied cathodic bias[24].

and 356 cm⁻¹)[25] were absent. We identified three bands that are not present in dry histidine samples without electrochemical bias: 1009, 1640, and 2079 cm⁻¹ indicating possible new interactions between histidine with either reduced Cu substrate, CO₂ (or related intermediates), or the electrolyte under applied cathodic biases. The band around 2079 cm⁻¹ is particularly interesting, as it is typically assigned to C≡O stretching during in-situ Raman on Cu[25]. However, a similarly positioned band was observed on bare Cu and electrodeposited Cu₂O in the presence of 0.025 M dissolved histidine (Fig. S5.3a, b) and Cu-Hist samples (Fig. S5.3c) in N₂ purged 0.1 M KHCO₃. Thus, we ascribe the band around 2079 cm⁻¹ to histidine-related interactions with Cu or the electrolyte under cathodic potentials, and not to the typical C≡O stretching from adsorbed *CO on Cu (* denotes adsorption site).

The in-situ Raman result on electrodeposited Cu₂O in presence of dissolved histidine in KHCO₃ electrolyte is fascinating because it suggests that histidine molecules can adsorb, interact, and persist on catalyst surface during CO₂RR-relevant potentials. Histidine is a unique molecule that has different forms depending on the degree of protonation. In the bulk electrolyte (CO₂ saturated 0.1 M KHCO₃, pH ≈ 6.8), histidine is expected to be in a mixed His⁺/His± state[26], allowing it to be attracted to the cathode. More importantly, it can react with CO₂ to form a zwitterion carbamate[27], allowing the more positively charged imidazole ring to still approach the cathode from the double layer for subsequent interaction. Further explanation of histidine's attraction towards the cathode and interaction with the Cu surface is described in SI Sections 5.2 and 5.3.

We continued the in-situ Raman investigation on Cu-0 in CO₂-purged 0.1 M KHCO₃ electrolyte (Fig. 3b). As expected, Cu-0 behaves just like typical Cu₂O-derived Cu catalysts[28], where the Cu₂O bands disappear almost instantly when −0.10 V cathodic potential was applied. The expected C≡O frustrated rotation and Cu-CO bands at 279 and 364 cm⁻¹ were also observed clearly once the potential reaches −0.7 V onwards, indicating the suitability of our system to detect the signature of such intermediate species. We note that similar bands are also observed on benchmark measurement on

electrodeposited Cu₂O in CO₂-purged 0.1 M KHCO₃ electrolytes (Fig. S5.3d).

We then focused on Cu-Hist as the sample with the best CO₂RR activity and selectivity in this study. Only Cu₂O-related bands at 519 and 620 cm⁻¹, and broad humps around 1130 cm⁻¹ were present on unreduced Cu-Hist at OCP (Fig. 3c). The absence of histidine-related bands at OCP is reasonable, as the initial interaction of histidine on unreduced Cu-Hist is expected to be a mixture of physical and chemical (see Fig. S2.6 and SI Section 5.3 for details). The coverage of chemically bonded histidine on unreduced Cu-Hist was relatively low, with approximately one histidine molecule per 16−26 surface Cu atoms as inferred by XPS (Figs. S2.8 and S2.9). At −0.10 V cathodic voltage, Cu₂O-related bands immediately disappeared, accompanied by the appearance of CO₃²⁻ band (1073 cm⁻¹) and a series of strong bands that closely matched dissolved histidine experiment and literature values (major: 1009, 1259, 1321, 1485, 1572, 1640, 2081 cm⁻¹). These histidine-related bands on Cu-Hist were markedly more intense than the ones observed during dissolved histidine experiments (Fig. 3a), even though the effective histidine concentration in the system should be much lower. The expected Raman bands related to Cu−CO and C≡O frustrated rotation (expected around 279 and 364 cm⁻¹) were also missing in Cu-Hist, even after ramping the cathodic potential to −1.10 V.

We posit that the initial strong chemical and physical interaction between histidine and Cu₂O in the unreduced stage through Cu−N bonding (inferred from additional Cu 2p peak at higher binding energy and significantly shifted N 1s of Cu-Hist sample, Fig. S2.9) may be critical in achieving high surface coverage of histidine during catalysis. We observed superior CO₂RR on Cu-Hist when compared to physically mixed histidine of similar loading (Fig. S4.6a). The missing Cu−CO and C≡O frustrated rotation (expected around 279 and 364 cm⁻¹) on Cu-Hist, even at very cathodic potential is intriguing. Given the excellent C₂₊ selectivity on Cu-Hist, the persistent histidine Raman bands and missing M-CO bands at very highly cathodic potentials indicate that strongly adsorbed histidine might have altered the interactions

**(a)**

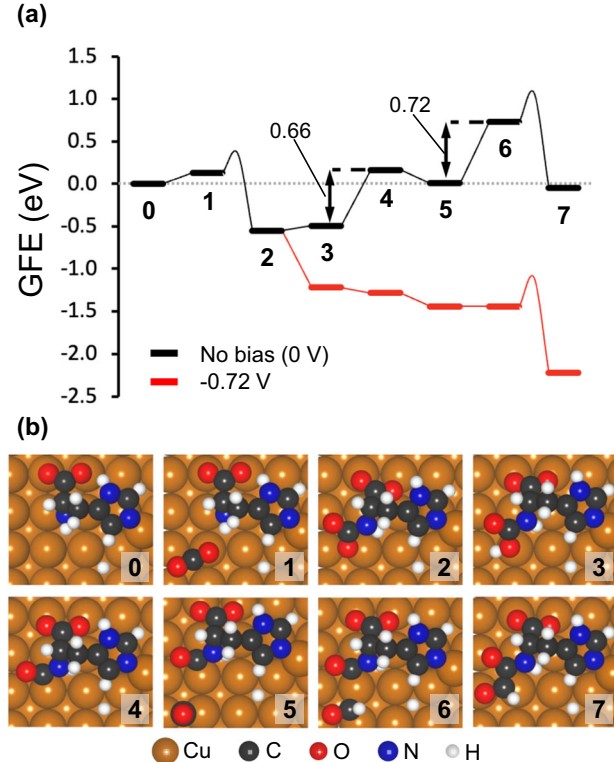

**(b)**

Cu ● C ● O ● N ○ H

**Fig. 4 | Initial reaction steps during $CO_2RR$ over histidine-Cu/Cu(100) substrate calculated by DFT. a** Gibbs free energy (GFE) diagram and the **b** snapshots of the first few surfaces intermediates in the histidine-assisted $CO_2RR$ mechanism. The GFE diagram was calculated from the reference state (**0**), which consists of a histidine-Cu/Cu(100) shown in Fig. S6.2b, a gas-phase $CO_2$ molecule, and an adsorbed *CO. In configurations (**1–4**), the *CO molecule adsorbed on a bare Cu(100) substrate is omitted for clarity, however, the energy has been added to each system accordingly. This *CO approaches the Hist−CO intermediate (**5**) and becomes *CHO after a CPET step (**5→6**), where the thermodynamic barrier $\Delta G_{5→6}$ of 0.72 eV can be overcome with applied bias (red line). The newly formed *CHO species may couple with the co-adsorbed Hist−CO intermediate through a C−C bond to form **7**. The intermediates of the surface reactions (**1→2** and **6→7**) are connected by smooth lines, from which the energy level of the TS may be inferred (the highest point of the smooth lines).

between Cu and *CO (or related intermediates). Previous reports of amino group interactions with $CO_2RR$ intermediates have been confined only to weak interactions[29], although altered *CO surface coordination to Ag surfaces in the presence of amino-containing triazole has been reported[30].

**Theoretical calculations**

We then turn to DFT calculations to rationalise the effects of histidine on the selectivity towards $CH_4$ and $C_2H_4$ over the Cu-Hist catalyst. A single deprotonated histidine molecule was placed over a 4 × 4 Cu(100) surface slab, based on the observed state and coverage of histidine in our synthesised catalyst from the XPS (SI Section 2.5) and Raman data (Fig. 3c and SI Section 5). Details of the surface model optimisation are described in SI Section 6.2. It is widely accepted that the $CO_2RR$ on Cu proceeds through a common *CO intermediate[31]. However, our experimental results indicate a possible deviation from the typical *CO−catalyst interaction (absence of Cu−CO and C≡O frustrated rotation). Accordingly, we explored an alternative *$CO_2$ to *CO conversion with subsequent transformation into $CH_4$ and $C_2H_4$ with direct involvement of a histidine molecule. In the following discussion, all intermediates during the reactions are labelled with bold numerals, while each elementary step is represented by A→B, where (A) and (B) are two

consecutive intermediates. In the description of reaction intermediates, the "Hist" label refers to the co-adsorbed histidine molecule. We discuss the thermodynamics of the transformation in terms of Gibbs free energies (G), and Gibbs free energy change ($\Delta G_{A→B}$).

First, a $CO_2$ molecule approaches the surface and physisorbs on Cu sites near the deprotonated amine group from histidine (**1**, Fig. 4). The *$CO_2$ adsorbate may bind the N atom in histidine (**1→2**) by overcoming a barrier of 0.23 eV to form the Hist−$CO_2$ complex (**2**). The C−N coupling is highly exergonic with $\Delta G_{1→2}$ of −0.68 eV, indicating a thermodynamically favoured product. The electrochemical conversion of Hist−$CO_2$ to Hist−CO involves two coupled proton−electron transfer (CPET) steps. The first CPET forms Hist−COOH (**3**) in a slightly endergonic process ($\Delta G_{2→3}$ = 0.05 eV), while a subsequent CPET generates $H_2O$ and Hist−CO intermediate (**4**) on the surface ($\Delta G_{3→4}$ = 0.66 eV). From here, a surface *CO, originally on Cu sites distant from the histidine molecule, approaches to the Hist−CO intermediate (**5**). The free energy change ($\Delta G_{4→5}$ = −0.16 eV) suggests that the approach of *CO to sites near Hist−CO may occur spontaneously.

A following CPET would transform the *CO into *CHO (**6**) with $\Delta G_{5→6}$ of 0.72 eV, which is the most endergonic step where a modest applied potential should be applied to make the reaction proceed (Fig. 4a, red curve). Once the generated *CHO (thermodynamically favoured over *COH on Cu(100)[32]) is present around the Hist−CO intermediate, the C−C coupling between the histidine-bound *CO and *CHO (**6→7**, Fig. 4b) is both kinetically ($E_a$ = 0.33 eV) and thermodynamically ($\Delta G_{3→4}$ = −0.78 eV) more favourable than the baseline cases (*CO-*CO and *CO-*CHO coupling on Cu(100) surface, SI Section 6.8). In our calculations, the *CO-*CO coupling over Cu(100) is endergonic by 0.96 eV and has a free energy barrier of 1.31 eV (Fig. S6.8), while the coupling between *CO and *CHO is endergonic by 0.10 eV and has a barrier of 0.63 eV.

Apart from the *CO and *CHO adsorbed on the surface, other C−C coupling alternatives from (**5**) were also explored (e.g., coupling between Hist-CO and *CO or Hist-CHO and *CO). However, the resulting activation energies were found to be considerably higher (1.13 and 1.43 eV, SI Section S6.9) than the (**6 → 7**) coupling. In addition, once the Hist-CO-CHO (**7**) is formed, it may be transformed into Hist-COH-CHO, Hist-CHO-CHO, Hist-CO-CHOH, or Hist-CO-$CH_2O$ during the following CPET (SI Section S6.9). We found that the reaction is most likely to proceed via Hist-CO-$CH_2O$ (**8**, Fig. 5a) due to comparatively higher thermodynamic barriers in other reaction channels.

In Fig. 5a, we can see all CPET steps following (**7**) are not potential limiting, as most of them are downhill steps and the $\Delta G$ for uphill steps (**11→12**) and (**13→14**) are <0.72 eV ($\Delta G_{5→6}$). After the Hist−$CH_2CH_2$ intermediate (**14**) is formed, it needs to be decoupled from the histidine fragment to complete the whole reaction. This step requires overcoming an energy barrier of 0.79 eV to break the C−N bond (**14→15**). However, this C−N bond cleavage (**14→15**) has a much lower energy level compared to the C−C bond coupling (**6→7**) (Fig. 5a), thus it should not be rate limiting. We found that the Hist−CO and *CHO coupling (**6→7**) has the highest energy level in the energy profile, which is the key step at 0 V (Fig. 5a). However, the $CO_2$ binding with Cu-Hist (**1→2**) becomes more important when a potential of −0.72 V is applied (Fig. S6.3).

As $CH_4$ formation is suppressed on Cu-Hist, a reaction pathway producing methane was also studied from (**4**) (Fig. 5b, grey shaded substrate and SI section S6.7) to understand the reason behind this. We found this pathway requires a surface reaction (C−N bond-breaking, **18→19**) to release $CH_2O$ from adsorbed histidine for the subsequent C protonation to produce $CH_4$. This step has an energy barrier of 0.99 eV, making it the rate-limiting step in the formation of $CH_4$ on Cu-Hist, and higher than the barrier of 0.57 eV for the rate-limiting (*CO→*CHO) on bare Cu (100) (Fig. S6.6).

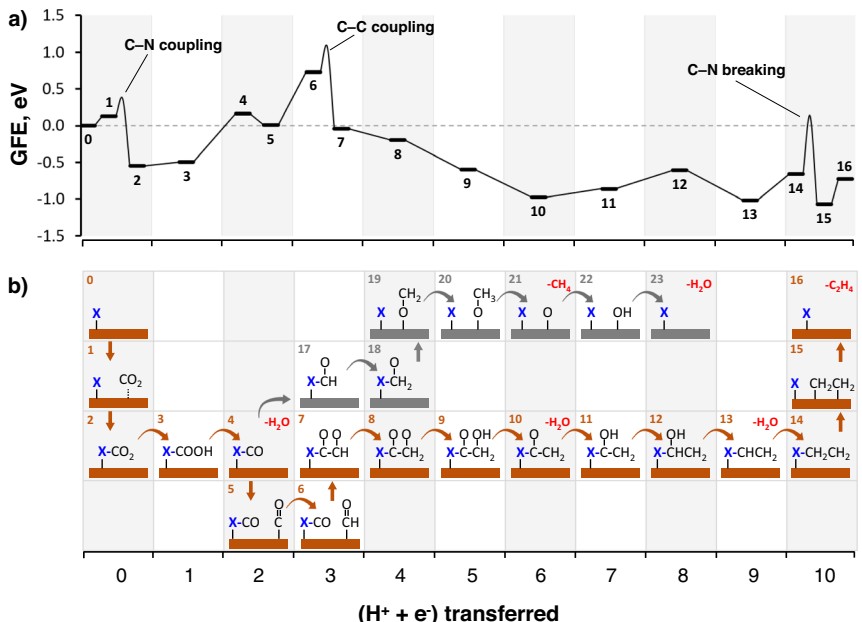

**Fig. 5 | Mechanistic understanding of favourable and unfavourable CO₂RR pathways on Cu-Hist calculated by DFT. a** Gibbs free energy diagram for $CO_2$ to $C_2H_4$ through the histidine-assisted mechanism. Configuration (0) is a reference configuration consisting of a histidine-Cu/Cu(100) (Fig. S6.2b), a gas-phase $CO_2$, and an adsorbed *CO. The intermediates participating in surface reactions (i.e., C−C coupling and C−N breaking steps) are indicated at the same abscissa and are linked by a curve illustrating the energy barrier of the process. The effect of applied bias on the free energy of this transformation is shown in Fig. S6.3. **b** Reaction pathways for the histidine-assisted electroreduction of $CO_2$ to $CH_4$ and $C_2H_4$ on Cu(100). Histidine molecule is represented by a blue "X". Note that the reaction may follow two separate pathways from (4), through either (5) towards $C_2H_4$ (marked with orange substrates) or (17) towards $CH_4$ (marked with grey substrates). Details on the histidine-assisted formation of $CH_4$ are discussed in SI Section 6.7. Desorbed molecules during the chemical process are indicated in red. The number at the top left corner of each box represents the reaction step number as described in the main text.

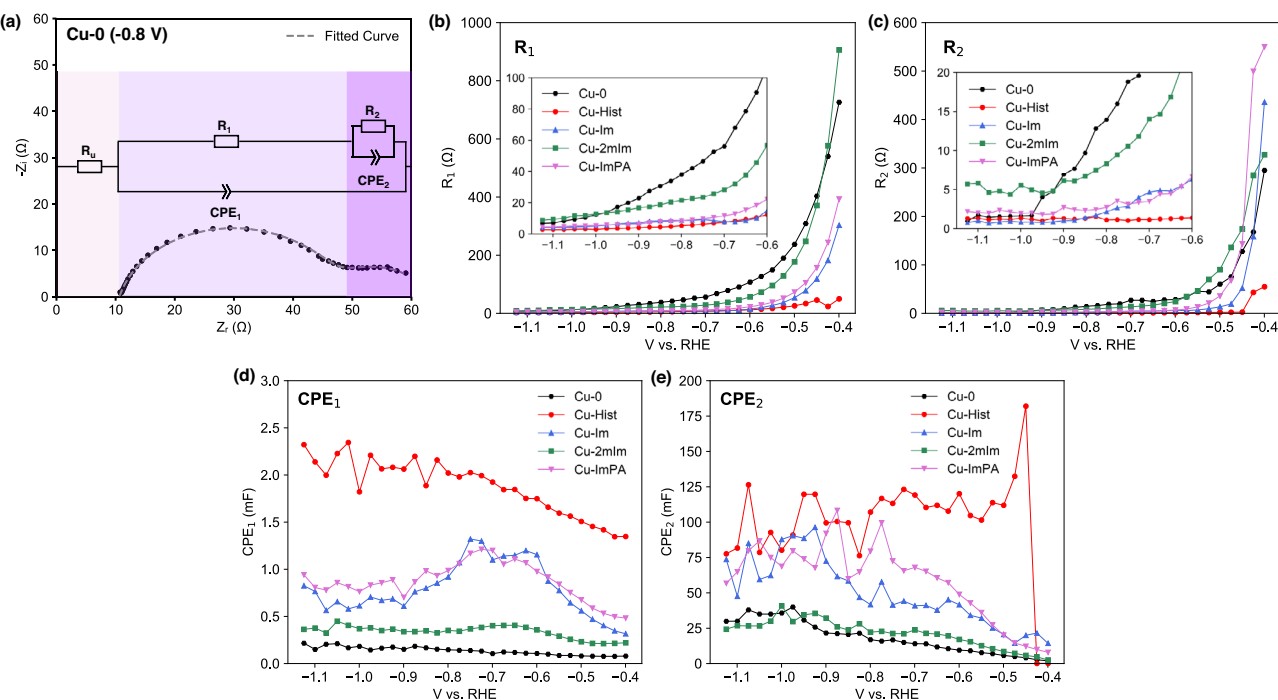

**Fig. 6 | Evaluation of organic functionalised Cu catalysts using EIS measurement at catalytically relevant DC potentials. a** The equivalent circuit used for EIS spectra fitting. Numerical data fitting results of EIS spectra measured on five samples across different DC potentials: **b** $R_1$, **c** $R_2$, **d** $CPE_1$, and **e** $CPE_2$.

Ergo, the presence of histidine plays opposite roles for $C_2H_4$ and $CH_4$ formation. In addition, the limited availability of H⁺ near the surface (due to the expected basic pH in the double layer) and the increased concentration of *CO species near histidine may also account for the more favourable $C_2$ product pathway.

This alternative mechanism for CO₂RR via histidine-assisted transformations may help rationalise the absence of the C≡O frustrated rotation in the Raman bands at applied bias during our experiments. On the one hand, *CO₂ may be transformed into *CO while bound to the histidine molecule through the amine N atom

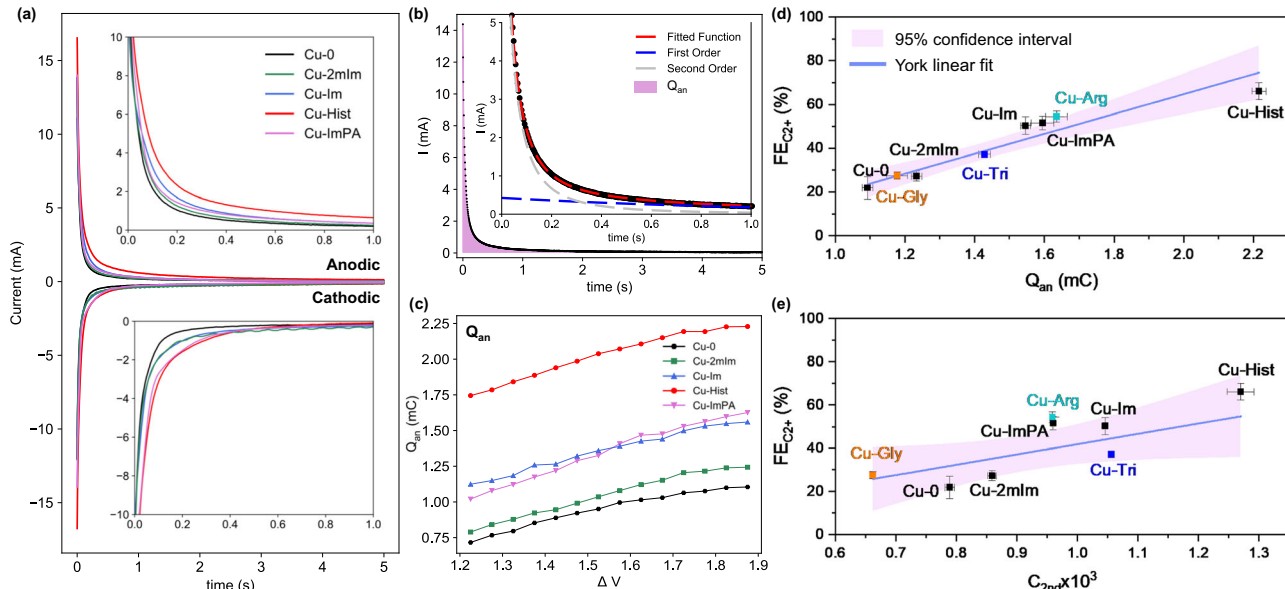

**Fig. 7 | Evaluation of organic functionalised Cu catalysts using mPV measurement at catalysis-relevant DC potentials. a** The transient current responses of Cu-0 and four organic functionalised samples at $\Delta V = 1.8$ V, showing transient anodic and cathodic current responses immediately after the applied voltage pulse. **b** Detailed view of transient anodic pulse for Cu-0 at $\Delta V = 1.8$. Shaded area represents the integrated transient anodic current decay ($Q_{an}$). Broken blue and grey lines represent the 1st and 2nd order decay equation fitting respectively, and the composite function is represented by broken red lines. Insets in **a** and **b** show enlarged portion of the current decays up to the first second. **c** integrated $Q_{an}$ value of Cu-0, Cu-2mIm, Cu-Im, Cu-Hist, and Cu-ImPA with increasing $\Delta V$. **d**–**f** Correlation plot between average **d** $Q_{an}$, **e** $C_{2nd}$ and the corresponding $C_{2+}$ product selectivity (FE$_{C2+}$) at −1.6 V. Orange, blue and turquoise coloured data points are additional validation points of Cu-Gly, Cu-Tri, and Cu-Arg. Error bars represent the standard deviation of three independent measurements. Solid purple lines and shaded areas in **d** and **e** represent York linear fit (considers both x- and y-errors), and the 95% confidence interval band, respectively.

($1 \rightarrow 2 \rightarrow 3 \rightarrow 4$), limiting the interaction of *CO with the Cu surface sites. On the other hand, the estimated high surface coverage of histidine (1 molecule per 16−26 surface Cu) may limit the amount of surface *CO bound to Cu sites, considerably affecting the intensity of characteristic Raman bands. In addition, the resulting few Cu−*CO intermediates may transform quickly into *CHO at high negative potentials (<−0.72 V) according to the Boltzmann probability distributions (SI Section 6.5 and Fig. S6.4), reducing further the measurable Cu−*CO indicators.

**Electrochemical surface charge investigations**

From in-situ Raman and DFT investigations, we learnt that histidine remains specifically adsorbed on Cu-Hist surface during CO₂RR and can provide alternative pathways towards C₂₊ via the formation of Hist-CO intermediate. The accumulation of such surface-adsorbed species can change the local electric field and provide additional electrostatic interaction near the catalyst surface, akin to the cation effect reported previously[33–35]. Therefore, we enlist transient electrochemical techniques in an attempt to quantify the changes in the surface charge. At the same time, we posit that the metrics obtained by transient electrochemical techniques may also be exploited as "activity descriptors" to predict CO₂RR activity over a wide range of catalyst systems.

We first turn to EIS as the established technique capable of probing time-dependent reaction mechanisms[36], from which $R_{CT}$ and capacitance values can be extracted and linked to catalytic activity. However, EIS measurements in literature are often performed at potentials irrelevant to the catalysis process, assuming that the reaction progression is governed by electron transfer kinetics[37]. Therefore, we performed EIS across a wide cathodic potential range from −0.400 to −1.125 V under CO₂RR conditions (CO₂ purged 1 M KHCO₃). A modified Randle's circuit model that accounts for two possible electrochemical interfaces with different timescales was used (Fig. 6a), in line with the observation of a second semicircle at lower frequencies and more cathodic potentials (SI Section 7)[37,38]. The extracted $R_{CT}$ values at different time domains (labelled $R_1$ and $R_2$) were then compared across a wide applied cathodic voltage. Here, $R_1$ corresponds to

the high-frequency component (>50 Hz, Fig. 6b) and $R_2$ the low-frequency (<50 Hz, Fig. 6c). $R_1$ is frequently attributed to the double layer component, while $R_2$ is assigned to the slower Faradaic or pseudocapacitive element due to the more relevant timescale[38,39]. Both $R_1$ and $R_2$ change inversely to applied cathodic voltage and converge to similar values on all samples, indicating that electron transfer pathways are generally facile during CO₂RR. Functionalised samples do show lower $R$ values compared to Cu-0, but the differences are generally very small (<9 Ω for $R_1$ and <5 Ω for $R_2$) once the applied potential reaches CO₂RR relevant potentials (<−1.0 V).

The capacitive terms were also assessed, as they can be closely related to accumulated species at or near the cathode surface[36]. This term is represented by a constant-phase element (CPE) reflecting the frequency dispersion behaviour of the solid-electrolyte interface[39]. Like the R terms, the CPE component consists of CPE₁ and CPE₂ representing the higher and lower frequency regions respectively. A clearer divergence between the CPE values of functionalised and unfunctionalized surfaces was prevalent−the most active Cu-Hist displaying ≈13.9× and ≈5.5× higher average CPE₁ and CPE₂, respectively, than Cu-0 ($V < $−1.0 V), an indication of higher population of charged species in the double layer or near catalyst surface at the onset of CO₂RR that may be related to the adsorbed histidine and intermediates.

At first glance, the general trend of $R_1$, CPE₁, and CPE₂ values at CO₂RR relevant potentials of −1.000 to −1.125 V appears to be in line with the observed CO₂RR trend (Fig. S9.1). However, large fluctuations at more cathodic potentials may originate from the momentary changes in capacitance values arising from vigorous gas evolution (Fig. 6d, e), hindering data collection for potentials more cathodic than −1.125 V due to vigorous bubbling (Fig. S7.3). Hence, we opine that EIS-derived metrics may not be suitable to describe CO₂RR activity due to large errors at catalytically relevant potentials.

In search of a better way to measure surface charge, we turn to a modified pulsed voltammetry (mPV) technique (see SI Section 8 for details). The mPV excitation pulse was constructed with a fixed upper

bound near the OCP (typically around +0.2 to +0.3 V) and gradually more cathodic lower bound voltage between around −0.5 to −1.6 V which is relevant to the CO2RR operating voltage of our catalysts. The applied mPV would trigger a repeating cycle of charged species accumulation (adsorption) during cathodic lower bound voltage and decumulation (desorption) at OCP. Like EIS and other electrochemical techniques[36,40–42], we recognise that mPV alone cannot ascertain the identity of adsorbed intermediates nor their adsorption behaviour. However, it serves to provide an estimate of the relative quantity and desorption kinetics of such species when the applied cathodic potential is removed.

Overall, the transient anodic and cathodic responses appear to grow with more cathodic lower bound on all samples (Fig. S8.1), indicative of higher surface charge density at more cathodic potentials as expected. The magnitudes of anodic and cathodic pulse decays of all samples at $\Delta V = 1.8$ V (approx. −1.5 to −1.6 V) are roughly balanced after subtracting the steady-state catalytic current (Fig. 7a), suggesting the reversibility of the charges. Due to the convolution of the catalytic current with the cathodic decay profile, the anodic decay profile was instead selected for further analysis.

We start by integrating the anodic transient response for 5 s (Fig. 7b). The integrated anodic transient charge ($Q_{an}$) grows almost monotonously with more cathodic lower voltage bound on all samples (Fig. 7c). This suggests a gradual build-up of charged species concentration on the catalyst surface with increasingly cathodic potentials. The rate of $Q_{an}$ growth with respect to $\Delta V$ was similar among the different samples, and the $Q_{an}$ trend of Cu-Hist > Cu-Im ≈ Cu-ImPA > Cu-2mIm > Cu-0 was practically maintained throughout the entire $\Delta V$ range. Further mathematical fitting and analyses (SI Section 8.1) found that the anodic transient current decay cannot be fitted to just one decay model commonly observed in electrochemical systems[43]. Instead, the best fit can be obtained with a combination of second-order and first-order decay kinetic functions (Fig. 7b), suggesting a convolution of at least two surface processes:

$$I = k_{2nd}\left(\frac{1}{C_{2nd}} + k_{2nd}t\right)^{-2} + C_{1st}k_{1st}e^{-k_{1st}t}$$

The second-order decay kinetic dominates early up to $t \approx 0.4–0.5$ s, which is ascribed to the cumulative desorption of species on the catalyst surface. The first-order decay term dominates at the longer timescale of $t > 0.5$ s, representing slow background processes occurring near OCP, such as Cu re-oxidation (SI Section 8.2). We focus on the second-order term and extract the $C_{2nd}$ and $k_{2nd}$ coefficients representing the initial surface charge accumulation and desorption rate respectively.

The extracted $C_{2nd}$ values trend follows: Cu-Hist > Cu-Im > Cu-ImPA > Cu-2mIm > Cu-0, which is in the same order as the CO2RR activity to C$_{2+}$ product. Meanwhile, $k_{2nd}$ trend is inversed (Cu-0 > Cu-2mIm ≈ Cu-ImPA > Cu-Im ≥ Cu-Hist). Thus, higher CO2RR catalytic activity is linked to higher initial surface charge accumulation (high $C_{2nd}$) and slower desorption (low $k_{2nd}$), which leads us to hypothesise that superior CO2RR performance to C$_{2+}$ product is related to better intermediate stabilisation aided by the organic functionalisations. We found that the trend of mPV derived parameters ($Q_{an}$, $C_{2nd}$, and $k_{2nd}$) are still in line with the C$_{2+}$ product selectivity trend, even with the inclusion of other organic molecules where the imidazole group are absent, namely arginine, glycine, and 1,2,3-triazole (Cu-Arg, Cu-Gly, Cu-Tri, Figs. S4.7, S7.4, and S8.6). Ergo, this correlation suggests that mPV may be used to predict C$_{2+}$ selectivity on wider types of catalyst surfaces.

Overall, both EIS and mPV measurements reflect elevated surface charge on organic-functionalised surfaces in the same order of CO2RR to C$_{2+}$ activity compared to bare Cu-0. Among all parameters derived from EIS and mPV, $Q_{an}$ stands out as a simple yet elegant choice as a

catalytic activity descriptor, as a more accurate integration of $Q_{an}$ can be obtained across different samples at CO2RR-relevant potentials. The correlation between $Q_{an}$ and FE$_{C2+}$ (Fig. 7d, also $j_{C2+}$ in Fig. S8.2b) predicts that catalysts with a higher population of charged species on the surface potentially produce more C$_{2+}$ products, which may be related to the effectiveness of the surface functionalisation in stabilising CO2RR intermediates. We speculate that the concentration of charged species on the catalyst surface during CO2RR may be more important than the stability of these species at OCP, as inferred from the better correlation of $C_{2nd}$ with FE$_{C2+}$ compared to $k_{2nd}$ (Fig. 7e, also $j_{C2+}$ in Fig. S8.2d).

Parallels can be drawn between our results that show clear CO2RR activity modulation in presence of histidine to the cation effect on CO2RR selectivity[33–35]. A major difference between the cation effect and our Cu-Hist is that histidine can be specifically adsorbed on Cu$_2$O-derived Cu surface under cathodic bias, as shown by the in-situ Raman data (Figs. 3 and S5.3). Further, the hydration shell of histidine (and possibly other amino acids) is much larger than alkali cations, but softer[44], thus enabling higher surface charge accumulation than Cs$^+$ cation while allowing for specific adsorption to reduced Cu surface. In addition, much stronger interaction with intermediate is afforded on Cu-Hist through the amine group, unlike cations where only electrostatic effects are available[35]. To increase the surface charge, we speculate that the amino acids could be modified by placing electron-donating groups on the amino nitrogen to increase basicity, whilst balancing the interaction with *CO or *CHO intermediate coupling. One could also combine the effect of cation together by swapping K$^+$ with Cs$^+$ which may allow further stabilisation of the intermediate through additional electrostatic effect.

Our work highlights the effectiveness of organic surface functionalisation in enhancing the CO2RR activity and tuning the selectivity of transition metal catalysts. Particularly, histidine's unique combination of functional groups allows for the stabilisation of CO2RR intermediates and robust anchoring to the catalyst surface. When combined with Cu$_2$O-derived Cu, Cu-Hist sample displays significantly higher C$_{2+}$ across a very wide voltage range of >1 V, with up to 76.6% FE$_{C2+}$, with its overall performance and stability outperforming recent C$_{2+}$-selective catalysts (SI Section 10). In-situ Raman underlines strong interaction between histidine and Cu that persists at CO2RR relevant potentials, which may stem from the close interaction between Cu$_2$O and histidine in the catalyst precursor before reduction. The stabilised histidine on the Cu surface appears to allow alternative CO2RR pathways through direct interaction of *CO$_2$ with histidine, as shown by DFT calculation. Under negative potentials, the formation of C$_{2+}$ products is preferred over methane on the Cu-Hist catalyst due to lower energy barriers in the key reaction steps.

More interestingly, we discover that the stabilised organic functional groups increased the surface charge near the catalyst surface, as determined by EIS and mPV measurements. While the surface charge modulation effects may be similar to the cation effect, the organic functional groups employed in our case are likely to be specifically adsorbed, providing a more stable and potent means to boost CO2RR selectivity by opening new pathways for intermediates stabilisation.

We also found that the CO2RR enhancement of the organic-functionalised catalyst is highly correlated to the catalyst surface charge measured using the mPV method. Additional measurements on seven other molecules demonstrate that such correlations can be generalised beyond histidine and imidazole-containing functionalisation. Our results highlight the potential of surface charge measurements as a powerful addition to the characterisation arsenal for electrocatalytic activity[45]. With further validation works on other catalytic systems and creating more accurate regression models, we expect surface charge measurements to be a very useful proxy for future catalyst discovery, including organic−inorganic hybrids.

## Methods

Functionalised $Cu_2O$ was synthesised using a simple and scalable wet synthesis method. In brief, stoichiometric amounts of D-glucose (5 mmol, 0.90 g), anhydrous copper chloride (10 mmol, 1.34 g), and 1.5 mol.% (0.15 mmol) of the molecule of interest were dissolved in 50 mL of deionised water. The solution was heated to 75 °C in a water bath and hydroxide ions were introduced into the reaction mixture in excess via the dropwise addition and 20 mL of 2 M sodium hydroxide solution. The reaction mixture was left to stir for 1 h and the mixture was subsequently centrifuged at 6000 rpm (4830×g) for 4 min. The precipitate was washed and dried overnight at 60 °C in a vacuum oven. More details on materials and synthesis are described in SI Section 1.

XPS data were acquired at HarwellXPS. A Kratos Axis SUPRA was used, employing monochromatized Al kα (1486.69 eV) X-rays at 15 mA emission and 12 kV HT (180 W) and a spot size/analysis area of $700 \times 300 \, \mu m$. Gas cluster ion source is employed to perform the depth profile of the Cu-Hist. Details on XPS measurements are presented in SI Section 2.5. Additional materials characterisations are presented in SI Section 2.

$CO_2RR$ experiments were performed in a custom electrochemical H-cell made of PEEK and PTFE. The electrolyte compartments were separated by an anion exchange membrane and each electrolyte compartment was filled with 8 mL of electrolyte. Gas products were quantified using gas chromatography and liquid products were quantified using $^1H$ NMR. Additional H-cell measurements, including baseline, control experiments with physically mixed histidine–$Cu_2O$ or glassy carbon, and calibration data are presented in SI Section 3.

An open cell without a membrane was used for mPV and EIS measurements. Electrochemical impedance spectroscopy (EIS) data were collected from pre-reduced catalysts with 1 M $KHCO_3$, with higher electrolyte concentration used to minimise the $R_u$ as well as create a more stable EIS environment. The spectrums were then collected across 30 different potentials between the range of −0.400 to −1.125 V vs. RHE at 0.025 V increments. The spectra were collected from 0.5 Hz to 30 kHz at 10 data points per decade on a Gamry 600+ potentiostat. The obtained Nyquist plots were individually fitted using the Simplex method on Gamry Echem Analyst (v7.9.0). Detailed experimental data of EIS experiments are presented in SI Section 4.

Raman spectroscopy was performed with a confocal Raman microscope (LabRAM HR Evolution, Horiba Jobin Yvon) in an epi-illumination mode (top-down) with He–Ne laser (633 nm, Pacific Lasertech) excitation source. A water immersion objective lens (LOMO APO water phase, ×70, numerical aperture: 1.23) protected by a 0.013 mm thin Teflon film was used to collect spectra. A custom-made PTFE open cell filled with $CO_2$ (or $N_2$) purged 0.1 M $KHCO_3$ was used to hold 10 mm electrode and perform the in-situ electrochemistry, controlled by a Gamry 600 potentiostat. Gas purging is maintained during measurement. More details on the in-situ Raman and additional measurements are presented in SI Section 5.

The DFT calculations were carried out with the projector augmented wave method (PAW)[46] and the PBE functional[47] with D3 dispersion corrections[48] as implemented in the VASP software[49–51]. An energy cut-off of 400 eV was used for the plane waves and the occupancies were treated with a Gaussian smearing technique with a width of 0.05 eV, but eventually extrapolated to zero width. We consider that a self-consistent field (SCF) cycle was converged when the change in energy was lower than $1 \times 10^{-5}$ eV, while the atomic optimisations were stopped when all forces were below $0.02 \, eV \, \text{Å}^{-1}$. All surface calculations were done with a $4 \times 4$ Cu(100) surface slab cell where the Brillouin zone was sampled with a $4 \times 4 \times 1$ Monkhorst–Pack mesh[52]. The NEB method[53] (and its climbing-image modification[54]) was used to locate the transition states and the finite differences method was used to approximate the harmonic vibrational frequencies. The Gibbs free energy for each intermediate and transition state was corrected with the corresponding terms (zero-point energy, thermal and entropy effects), including solvation. An extended description of the computational details, the surface model used in this work, energy corrections, and alternative pathways comparisons are presented in SI Section 6.

mPV experiments were conducted in 0.1 M $KHCO_3$, and the catalysts were pre-reduced at −1.125 V for 2000 s. The anodic potential was set around OCP, typically around +0.3 V, where Faradaic processes were minimal. The cathodic potential is varied at 0.05 V intervals, with $\Delta V$ values ($\Delta V = V_{anodic} - V_{cathodic}$) ranging from 0.775 to 1.875 V. Both anodic and cathodic pulses were applied for 20 s and the sampling time of the current was 0.004 s. The anodic current decays were integrated from $t = 0$ s to $t = 5$ s using scipy.integrate.simps while mathematical fitting of the pulse from $t = 0$ s to $t = 2$ s was conducted using the scipy.optimise.curve_fit function with positive bounds. Detailed experimental data of EIS, mPV, and all parameter correlations are presented in SI Section 7–9.

$CO_2RR$ activity, selectivity and stability performance benchmarking with literature is presented in SI Section 10.

## Data availability

The authors declare that all data supporting the results of this study are available within the paper and its supplementary information files. Product quantification, EIS, modified pulse voltammetry and variable rate cyclic voltammetry data is available from Figshare database under accession code (https://doi.org/10.6084/m9.figshare.21787079).

## Code availability

Modified Pulse Voltammetry data treatment code is available at Figshare database under accession code (https://doi.org/10.6084/m9.figshare.21787079).

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

## Acknowledgements

This work is supported by A*STAR (The Accelerated Catalyst Development Platform (A19E9a0103), Accelerated Materials Development for

Manufacturing (A1898b0043) and Career Development Award (202D800037)), as well as the National Research Foundation, Singapore, and A*STAR under its LCERFI program (Award No U2102d2002). J.M.A.-R., M.B.S., and J.Z. acknowledge the use of high-performance computational facilities from the National Supercomputing Centre (NSCC) Singapore (https://www.nscc.sg) and A*STAR Computational Resource Centre (A*CRC). B.S.Y. acknowledges support from Ministry of Education, Singapore (A-0004135-00-00). M.Y. acknowledges studentship from A*STAR Research Attachment Programme. The authors acknowledge Dr. Debbie Seng Hwee Leng, Dr. Zhang Mingsheng, Mr. Andrew Wong Jun Yao, Ms. Sng Anqi, and Dr. I Made Riko of Institute of Materials Research and Engineering for their assistance in performing initial XPS measurements, elemental analyses, and mathematical fitting discussions. We also thank Dr. Benjamin Chen of the Institute of High Performance Computing for useful discussions on the role of solvent in solid–liquid interfaces. Any opinions, findings and conclusions or recommendations expressed in this material are those of the author(s) and do not reflect the views of A*STAR.

## Author contributions

C.Y.J.L.: Synthesis, electrochemical data curation and analysis, analysis code writing, figure drawing and writing, M.Y.: Synthesis, electrochemical and materials data curation and analysis, writing, J.M.A.-R.: Computational data curation and analysis, figure drawing, writing, A.D.H.:Idea conception, characterisation data curation and analysis, lead write, supervising, W.J.T.: In-situ Raman data curation and analysis, Y.Z.: synthesis development, materials, Z.H.J.K.: Electrochemical data curation, M.L.: Electron microscopy data curation and analysis, M.I.: XPS data curation and analysis, T.L.D.T.: Synthesis, materials, chemistry mechanism and intermediate analysis, Y.B.: Synthesis, materials characterisation, C.K.N.: Data analysis, chemistry mechanism and intermediate analysis, B.S.Y.: Raman data analysis and supervising, G.S.: Materials characterisation data analysis and supervising, I.P.P.: Materials characterisation data analysis and supervising, K.H.: Materials data analysis and supervising, M.B.S.: Theoretical data analysis and supervising, J.Z.: Theoretical data analysis and supervising and Y.-F.L.: Materials data analysis, writing, supervising.

## Competing interests

The authors declare no competing interests.
