## [Peer review file · Nature Communications]

REVIEWER COMMENTS

Reviewer #1 (Remarks to the Author):

In this work by Handoko et al, a histidine functionalized Cu showed improved multi-carbon (C₂+) selectivity compared to pristine oxide-derived Cu. The authors used various experimental and computational tools to explain the observation and attempted to propose a universal activity descriptor based on surface charges to account for selectivity change. Overall, this research is novel to some extent, but does not merit publication in such high-quality journal as Nature Communications. Some detailed comments are below.

A, In-situ Raman spectroscopy:

1. What was the electrolyte for in-situ Raman spectroscopic studies? Was it 0.1 M NaOH (line 144, pg5) or 0.1 M KHCO₃ (saturated by CO₂, figure 3 caption)? It is likely the latter case based on context in lines 144-151. Please make it clearer.
2. Under the pH of electrolyte used in the in-situ Raman studies, the histidine molecule is likely negatively charged. This raises the question: why and how the histidine molecules approach the negatively charged working electrode?
3. What is the interaction between the histidine molecule and Cu substrate? Is it physio-adsorption or chemical adsorption? It seems to me that the authors tended to believe the latter case (XPS on pg8 of SI, and discussion in the main text, lines 122-127, pg 4). If so, Stark shift is expected. Further, identification/validation of the exact atoms that intact with Cu is possible and expected to be discussed.
4. Why the multiple bands (in the range of 100-1600 cm⁻¹) attributed by the authors to histidine were not observed at low applied potential range (until -1.1 V) for the Cu-hist sample under CO₂ conditions (figure 3c). If there were a strong interaction between histidine and Cu, these bands should have been observed regardless of applied potentials.
5. Following above question, interestingly, under N₂ conditions, above bands could be observed at much lower applied potentials (starting at -0.1 V). The authors need to explain this discrepancy.
6. In figure 3c, the bands attributed to copper oxides persisted until potentials as negative as -0.7 V, inconsistent with the authors' claim (lines 160-161, pg5) that "Cu₂O bands disappear almost instantly when -0.10 V cathodic potential was applied". This observation has been noted by the authors (lines 177-180, pg 6) but no explanation was given. I feel this is important, as the histidine may play a role to stabilize Cu₂O – at least at low overpotential range – which is believe by some researchers to enhance C₂+ product selectivity.
7. Above three questions raised my concerns about quality and reliability of the obtained in-situ Raman data and associated hypothesis on *CO adsorption (e.g., lines 156-157, pg5; lines 181-184, pg6).

8. I do not believe operando can be used to term the Raman spec study. It is at most an in-situ study – the test conditions for Raman spec were distinct from the conditions for the CO₂RR performance evaluation.

B, Other comments:

1. The Cu/N ratio and Cu-N interaction are based on characterisation of the pristine Cu₂O-histidine sample, which is different from the real catalyst (at least Cu₂O became Cu). Drawing any conclusions from such characterisation should be cautious.
2. *CO as the starting point for DFT calculations is not sound to me because: a) as this paper aimed to propose a new mechanism, it is reasonable to examine whether the histidine has any impact on elementary steps along the CO₂ to *CO coordinates, and b) no Raman signal for *CO was observed to support that *CO was even an intermediate in the Cu-Hist case.
3. Water solvent should be used and possible hydrogen-bond interactions should be considered in the DFT calculations, in particular, in the presence of additional molecule layer.
4. It was not explained by DFT calculations why ethanol selectivity was largely promoted in the experiment (lines 116-118, pg4, and figure 2).

Reviewer #2 (Remarks to the Author):

This manuscript studied organic-functionalized Cu with improved C₂+ products in electrochemical CO₂ reduction. The organics include histidine, imidazole, 2-78 methylimidazole, imidazolepropionic acid, arginine, triazole and glycine. Some characterizations were carried out, such as Operando Raman, DFT calculations, EIS, mPV, to explore the mechanism on the interface or catalyst/electrode surface. The decoration method is interesting, which realizes high and stable C₂+ product selectivity. However, some discussions and conclusions need careful deduction. The surface charge was considered to be universal activity descriptor to explain CO₂RR activity and proxy to catalyst development. The integrated anodic transient charge (Q_{an}), derived initial surface charge accumulation (C_{2nd}) and desorption rate (k_{2nd}), are all correlated to C₂+ selectivity. These parameters seem to obtain from kinetic functions (kinetic parameter). It is rough and limited to conclude. What kind of charge on surface, from what surface species? The organics functioned Cu and pristine Cu (Cu₀) are comparable to use surface charge descriptor.

On the other hand, the Operando Raman, DFT calculations, EIS, mPV were carried out separately, but a systematical discussion is highly encouraged, even partially. Are there any relationships among results from different characterization? To construct a universal descriptor is important but difficult. The scope

of application is especially important. I doubt the study and conclusion apply only to the catalyst system in this manuscript.

Reviewer #3 (Remarks to the Author):

The manuscript by Handoko et al reports on the mechanistic understanding of Cu/organic electrocatalysts towards the reduction of CO₂ to C₂+ products. Under normal circumstances I would consider this an extremely well-trodden territory as the electroreduction of CO₂ by Cu was first reported by Hori about 30 yrs ago and has become a very hot topic in the last decade. Such a conclusion would be premature as there are many aspects of this paper which bring a fresh take to an old story. The strengths of this paper are:

A, The careful exploration of multiple catalyst systems to determine one which is both selective towards C₂ species and stable (for at least 48hrs).

B, Unique analysis on the catalytic performance using time resolved electrochemical measurements to extract descriptors of reactivity.

C, The surface charge descriptor is in good accord with chemical intuition toward activity and could serve as a useful tool for future design principles.

I believe this paper could make a solid addition to nature comm. but a few things need to be strengthened first.

I would suggest the authors consider the following issues to improve the overall impact of their work:

A, Can the authors discuss the activity of their catalyst in terms of turn over frequencies (TOF) to better gauge them against thermal catalysts and project on the potential to operate at high current densities (ca A/cm²) which would be the requisite for utilization of this catalysts for practical deployment.

B, The theoretical approach uses standard approaches based on the computational hydrogen electrode as popularized by Norskov and co-workers. None the less the results could be more effectively presented as a catalytic cycle where the energetics (at zero and higher applied bias is included in for each step. This is particularly true to the data in figure 5 which is less accessible. Can the authors comment on the expected surface abundance of CO and Formyl groups. The latter of which their energetics indicates is in low abundance yet is being invoked in a surface reaction with X bound CO. Can the authors comment on the potential role of solvent and electrolyte which are not included but are known to change energetics (sometimes substantially).

C, In the conclusion section the authors need to stretch themselves and project on how their finding can be used to enhance reactivity. Can they discuss methods for boosting surface charge, ex increasing the basicity of the amine, changing the ion content in the solvent layer adjacent to the surface etc. I feel this is a sadly missed opportunity to impact an ongoing conversation that has been going for a long time.

REVIEWER COMMENTS

Our response to reviewers' questions is written in blue font. **Yellow highlighted text** marks major changes to the main text. No markings were added into SI.

SUMMARY OF RESPONSES TO REVIEWERS' QUESTIONS

Reviewer 1

Question	Response
Under the pH of electrolyte used in the in-situ Raman studies, the histidine molecule is likely negatively charged. This raises the question: why and how the histidine molecules approach the negatively charged working electrode?	1. At pH \approx6.8, histidine is expected to exist as a mixture of His⁺ and His[±] form, allowing some attraction from electrolyte bulk to the cathode surface.2. Histidine is known to interact with CO₂, forming zwitterion carbamate even at alkaline condition. The zwitterion formation allows more positively charged imidazole ring to still be attracted to the cathode surface.3. Solvent and electrolyte effect: accumulated cations on cathode could attract negatively charged side of histidine. Such layer would ameliorate the repulsion effect and help the transport of histidine across the double layer. New SI section 5.2 is added to address this question
What is the interaction between the histidine molecule and Cu substrate? Is it physio-adsorption or chemical adsorption?	On Cu₂O precursor 1. We expect two kinds of interactions: physical and chemical. Histidine is expected to encapsulate the Cu₂O crystals physically. Histidine is also shown via XPS to bound to Cu₂O surface chemically. On reduced Cu during CO₂RR, chemical interactions are proposed to be retained due to the following proposed explanations: 2. Proximity effect due to physical encapsulation. In this regard, histidine does not need to be attracted from the electrolyte bulk during the reduction process.3. Formation of Cu-histidine complex¹⁴ during cathodic voltage, which could subsequently latch on to the reduced surface and form a new active surface (as adsorbed histidine complex).4. Electrochemical attraction and interaction with CO₂/HCO₃⁻ system. We see new bands forming that are not previously seen on dry histidine/Cu-histidine without applied potential.5. DFT calculations demonstrate that the adsorption of histidine on (reduced) Cu substrate is energetically more favourable by -4.77 eV. New SI section 5.3 is added to address this question.

Why the multiple bands (in the range of 100-1600 cm^{-1}) attributed by the authors to histidine were not observed at low applied potential range (until -1.1 V) for the Cu-hist sample under CO_2 conditions (figure 3c). If there were a strong interaction between histidine and Cu, these bands should have been observed regardless of applied potentials.	We did not expect clear Raman signals on Cu-hist sample (before reduction at OCP) because  1. Most of the histidine should be physically bound/encapsulated over Cu_2O 2. There is low concentration chemically bound histidine on Cu_2O surface 3. Cu_2O is not an effective Raman enhancement surface. We have repeated multiple Raman measurements and as the electrochemical voltage is applied and Cu_2O is reduced to roughened Cu, Cu-histidine bands can be seen more clearly. Figure 3 and Figure S5.3 is revised to address this question.
Persistence of Cu_2O related to histidine, and validity of $^*\text{CO}$ disappearance	 1. We believe the persistence of Cu_2O to be spectator effect. After receiving feedback from Reviewer 1, we have repeated in-situ Raman multiple times and observed no clear evidence of persistent Cu_2O peaks. 2. None of the five repeats on Cu-Hist samples show any $^*\text{CO}$ adsorption band that is expected around 279 and 364 cm^{-1} on Cu surface. 3. We have shown on measurements on Cu-0, and baseline measurements on regular Cu_2O samples grown separately via electrodeposition that $^*\text{CO}$ can be detected on our Raman system consistently. 4. Thus, we are confident in our position that the presence of histidine alters intermediate adsorption on Cu surface and that $^*\text{CO}$ bands are not observed on systems containing histidine under CO_2 purged electrolyte and cathodic bias up to the voltage limitation of our Raman system at -1.1 V RHE. Figure 3 and Figure S5.3 is revised to address this question
$^*\text{CO}$ as the starting point for DFT calculations is not sound to me because: a) as this paper aimed to propose a new mechanism, it is reasonable to examine whether the histidine has any impact on elementary steps along the CO_2 to $^*\text{CO}$ coordinates, and b) no Raman signal for $^*\text{CO}$ was observed to support that $^*\text{CO}$ was even an intermediate in the Cu-Hist case.	 1. As suggested by the reviewer, we explored a reaction pathway of the CO_2 conversion in the presence of histidine. Indeed, the CO_2 can be stabilized by histidine (Hist) via formation of Hist-CO_2 complex ($\Delta G_{1-2} = -0.68 \text{ eV}$). 2. Starting from Hist-CO_2 complex, sequential proton-electron transfer steps lead to Hist-CO (4) via Hist-COOH (3), where the first CO is generated with the aid of histidine. 3. Although the adsorption of histidine lowers the chance of CO_2 adsorption on the Cu surface, it is still possible to happen during the reaction. The CO_2 decomposition on the Cu surface results in the second $^*\text{CO}$, which prefers to adsorb beside the histidine-CO complex ($\Delta G_{4-5} = -0.16 \text{ eV}$). 4. Further protonation of $^*\text{CO}$ (6) and the subsequent C-C coupling between the generated $^*\text{CHO}$ and the Hist-CO complex produce the C_{2+} species, i.e., the Hist-COCHO intermediate (7). Figure 4, 5, and Supporting Information Figure S6.3 and S6.4 is revised to address this question.

Water solvent should be used and possible hydrogen-bond interactions should be considered in the DFT calculations, in particular, in the presence of additional molecule layer	 1. In the revised work we adopted the widely used semi-empirical approach (presented as SE1, [reference 22-26 of the supporting information]) can adequately describe the solvation energy of the relevant intermediates. 2. This semi-empirical approach is based on calculations with explicit water molecules, deriving the solvation correction for *OH, *R-OH, and *CO (carbonyl-containing) surface species, thus H-bond interaction has been considered implicitly in this approach. Supporting information section 6.6 and Figure S6.5 is added to address this question
---	---

Reviewer 2

The integrated anodic transient charge (Q_{an}), derived initial surface charge accumulation (C_{2nd}) and desorption rate (k_{2nd}), are all correlated to C_{2+} selectivity. These parameters seem to obtain from kinetic functions (kinetic parameter). It is rough and limited to conclude. What kind of charge on surface, from what surface species? The organics functional Cu and pristine Cu (CuO) are comparable to use surface charge descriptor.	 1. We speculate that the surface charge measured here may be related to local electrostatic interactions within double layer and interface. Such interactions have been proposed to be the possible reason of shifting CO_2RR preference in presence different alkali cation in the electrolyte. 2. Unlike the cation effects, whose interaction has been shown to be confined to non-covalent, histidine is rather unique because it is known to form specific adsorption on Cu through either carboxylic or imidazole nitrogen. (shown through our in-situ Raman data) 3. However, we acknowledge that, like most other electrochemical techniques, our proposed mPV measurements are not able to identify the adsorbed species, whether they have specific adsorption to the surface or the adsorption behaviour. We developed it as a simple way to estimate the accumulation of all charged species in the double layer and on the catalyst surface that is related to the CO_2RR activity. Further work on extending the technique to other samples and conditions are under way. We added discussion points comparing different surface charge interactions in main text Page 12 and 14
--	--

Reviewer 3

Can the authors discuss the activity of their catalyst in terms of turn over frequencies (TOF) to better gauge them against thermal catalysts and project on the potential to operate at high current densities (ca A/cm^2) which would be the requisite for utilization of this catalysts for practical deployment.	TOF calculation is added in new Supporting Information Section 3.3 and discussed in the main text.
The results could be more effectively presented as a catalytic cycle where	We have now added Figure S6.3 which presents the effect of applied potential in the CO_2RR via the histidine-

the energetics (at zero and higher applied bias is included in for each step.	assisted mechanism in Figure 5. The applied potential changes the relative energy level of C-N coupling (1 to 2) and C-C coupling (6 to 7) on the energy profile, leading to a change of the key step during the chemical process.
Can the authors comment on the expected surface abundance of CO and Formyl groups. The latter of which their energetics indicates is in low abundance yet is being invoked in a surface reaction with X bound CO.	We have calculated the Boltzmann probability distributions of *CO and *CHO at various applied potentials, the ratio of probabilities depends only on their energy difference. (Figure S6.3).
Can the authors comment on the potential role of solvent and electrolyte which are not included but are known to change energetics (sometimes substantially).	We added Figure S6.5 and corresponding discussions in SI Section S6.6. The energies of all intermediates in Figure 4 and Figure 5 have been corrected by using semi-empirical approach (SE1)
Can they discuss methods for boosting surface charge, ex increasing the basicity of the amine, changing the ion content in the solvent layer adjacent to the surface etc. I feel this is a sadly missed opportunity to impact an ongoing conversation that has been going for a long time.	We agree that it is important to suggest methods to modulate the surface charge further. We see some parallel between previously reported cation effect to our work, histidine is much more beneficial because of its ability to be specifically adsorbed, with larger hydration shell but softer compared to large cations. Discussion is added in main text page 14.

Reviewer #1 (Remarks to the Author):

In this work by Handoko et al, a histidine functionalized Cu showed improved multi-carbon (C2+) selectivity compared to pristine oxide-derived Cu. The authors used various experimental and computational tools to explain the observation and attempted to propose a universal activity descriptor based on surface charges to account for selectivity change. Overall, this research is novel to some extent, but does not merit publication in such high-quality journal as Nature Communications. Some detailed comments are below.

We thank the reviewer for his/her time in reviewing our manuscript. We respect the reviewer's view on our work. With the additional data in this revised manuscript, we are confident that our work can advance the understanding of CO₂RR on hybrid catalysts and bringing us one step closer to the realisation of CO₂ utilisation to multi carbon products. Appended below is our detailed point by point reply to your concerns.

A, In-situ Raman spectroscopy:

1. What was the electrolyte for in-situ Raman spectroscopic studies? Was it 0.1 M NaOH (line 144, pg5) or 0.1 M KHCO₃ (saturated by CO₂, figure 3 caption)? It is likely the latter case based on context in lines 144-151. Please make it clearer.

The electrolyte for in-situ Raman spectroscopic studies is 0.1 M KHCO₃ saturated by CO₂. 0.1 M NaOH was the condition adopted by reference 24. The revised manuscript in page 5 (under Figure 2) has clarified this point.

A series of weaker bands around 1009, 1155, 1259, 1485, 1572 and 1640 cm⁻¹ that are consistent with Raman bands of deprotonated L-histidine adsorbed on Cu in the literature (0.1 M NaOH, -0.6 to -1.0 V vs Ag/AgCl)²⁴ were also observed.

2. Under the pH of electrolyte used in the in-situ Raman studies, the histidine molecule is likely negatively charged. This raises the question: why and how the histidine molecules approach the negatively charged working electrode?

The reviewer is correct in pointing out that histidine is likely to be partially deprotonated on the carboxylic group at the electrolyte pH used in the in-situ Raman studies. However, at pH ≈6.8, histidine is expected to exist as a mixture of His⁺ and His[±] form (see e.g., 10.1016/j.vibspec.2005.01.003), allowing some attraction from electrolyte bulk to the cathode surface. Further, histidine is known to interact with CO₂, forming zwitterion carbamate even at alkaline condition (see e.g., 10.1016/j.cej.2016.08.066). The zwitterion formation allows more positively charged imidazole ring to still be attracted to the cathode surface.

Moreover, the electrolyte used in CO₂RR also contain cations that accumulate near the electrode surface upon application of negative bias. These accumulated cations could also in turn attract negatively charged side of histidine. The concentration of alkali cations builds up upon cathodic bias application and are stabilised by repulsive charge among the positively charged cations and the solvent (hydration, see e.g., Ringe, S. *et al.* doi:10.1039/c9ee01341e). Such layer would ameliorate the repulsion effect and help the transport of histidine across the double layer.

We note that histidine adsorption on Cu has been reported earlier, also under cathodic bias (Martusevičius, S., et al. doi:10.1016/0924-2031(95)00025-9). At increasingly higher pH, histidine adsorption can still be observed under Raman under electrochemical cathodic bias.

We have added the following line to the main text page 6:

Histidine is a unique molecule that has different forms depending on the degree of protonation. At the bulk electrolyte (CO₂ saturated 0.1 M KHCO₃, pH ≈6.8), histidine is expected to be in a mixed His⁺/His[±] state,²⁶ allowing it to be attracted to the cathode. More importantly it can react with CO₂ to form a zwitterion carbamate,²⁷ allowing the more positively charged imidazole ring to still approach the cathode from the double layer for subsequent interaction. Further explanation on the histidine attraction towards cathode and interaction with Cu surface is described in **SI Section 5.2** and **5.3**.

We have also added a new Section 5.2 in the supporting information (page 25) to include these explanations.

Explanation on the attraction of histidine towards cathode upon electrochemical bias

Histidine is likely to be partially deprotonated on the carboxylic group at the electrolyte pH used in the in-situ Raman studies. At pH ≈6.8, histidine is expected to exist as a mixture of His⁺ and His[±] form,⁹ and this form still allows some attraction to the cathode from electrolyte bulk. Histidine is also known to interact with CO₂, forming zwitterion carbamate even at alkaline condition,¹⁰ allowing the more positively charged imidazole ring to be attracted to the cathode surface.

Moreover, the attraction of histidine from electrolyte bulk to the double layer can also be promoted by the cation accumulation upon application of negative bias. The concentration of alkali cation is known to build up upon cathodic bias application, and are stabilised by repulsive charge among the positively charged cations and the solvent.¹¹ Such layer would ameliorate the repulsion effect and help the transport of histidine across the double layer.

We note that histidine adsorption on Cu has been reported earlier, also under cathodic bias.¹² At increasingly higher pH, histidine adsorption can still be observed under Raman under electrochemical cathodic bias, albeit with some change on how the histidine binds to the surface.

3. What is the interaction between the histidine molecule and Cu substrate? Is it physio-adsorption or chemical adsorption? It seems to me that the authors tended to believe the latter case (XPS on pg8 of SI, and discussion in the main text, lines 122-127, pg 4). If so, Stark shift is expected. Further, identification/validation of the exact atoms that interact with Cu is possible and expected to be discussed.

The main sample (marked as Cu-Hist) is Cu₂O synthesised in presence of 1.5 mol% histidine in the precursor solution. We expect two kinds of interactions between histidine and Cu₂O precursor, physical and chemical. We expect the histidine to encapsulate the Cu₂O crystals physically. On the surface, our data shows histidine is bound to Cu₂O surface chemically.

Physical interaction between histidine and Cu₂O

The physical interaction between Cu₂O and histidine can be seen in the *ex-situ* FTIR spectra (**Error! Reference source not found.**). Weak bands that can be attributed to L-histidine was clearly observed on Cu-Hist at 1.5% loading. Stronger and clearer bands were observed on 10%, indicating thicker encapsulation and stronger physical interaction.

Figure S2.6: Ex-situ FTIR on pure L-histidine (black trace), Cu-Hist (1.5%, red trace), and Cu-Hist (10%, pink trace). The yellow shaded area marks the expected strongest peak position of histidine. Cu^{1+} peak (representing Cu_2O) around 620 cm^{-1} is marked.

Chemical interaction between histidine and Cu_2O

The chemical interaction between Cu_2O and histidine in Cu-Hist can be seen in the *ex-situ* XPS (Error! Reference source not found.). Here, clear additional Cu2p peaks that can be attributed to Cu-N bonds, and merged a and b N1s peaks indicating strong interaction between Cu and both *proxima*- and *tele*-N in the imidazole ring.

Our XPS measurements highlighted by the reviewers demonstrate clearly that histidine binds chemically to Cu_2O . This interaction is expected, as strong interaction between histidine and Cu or Cu_2O through the nitrogen group or carboxylic group has been documented widely elsewhere (example: Feyer, V. *et al. J. Phys. Chem. B* **112**, 13655-13660, (2008).; also, Wang, C. *et al. Appl. Surf. Sci.* **453**, 173-181, (2018).).

Figure S2.8: (a) Cu2p and (b) N1s XPS for Cu-Hist sample. Three depth profile scans based on gas cluster ion etching time were performed: surface (0 s), 30 s and 300 s etching. Dotted lines are guide to the eye. “a”, “b” and “c” label indicates the expected N1s binding energy of pure histidine as indicated in (c).

Interaction between histidine and Cu after reduction

From the evidence we gather in in-situ Raman experiments (Main **Figure 3** and **Figure S**), we strongly believe that histidine is retained on reduced Cu₂O-derived Cu surface after reduction. Such retention is possible due to the following proposed explanations:

- (1) Proximity effect due to physical encapsulation. We have established earlier that Cu₂O precursor is surrounded by histidine. In this regard, histidine does not need to be attracted from the electrolyte bulk during the reduction process.
- (2) Histidine is also known to form complex with Cu.¹⁴ We posit that such Cu-histidine complex can be easily formed during Cu₂O reduction, which could subsequently latch on to the reduced surface and form a new active surface (as Cu-histidine complex).
- (3) Electrochemical attraction and interaction with CO₂/HCO₃⁻ system. We see new bands forming that are not previously seen on dry histidine/Cu-histidine without applied potential.

To support this argument, we performed additional control experiments in **Figure S5.3**, comparing dry histidine powder, wet histidine on Cu surface, drop casted Cu-hist on open circuit, and with -0.1 V vs RHE under N₂ and CO₂ purging. Here, we see clear shift and/or broadening of the peaks when compared to dry histidine, suggesting new interaction when in contact with Cu upon cathodic bias application. The most interesting observation is the appearance of new Raman band near 1625 and 2080 cm⁻¹ upon application of electrochemical bias. Bands near this position usually attributed to ν -C=O and ν -C≡O modes on metal surfaces.¹⁴ However, the fact that this band is also present in N₂ purged experiment, we posit that it is a clear indication of Cu-histidine interaction upon application of cathodic bias.

Figure S5.4: Comparison between dry histidine, wet histidine (with 0.1 M KHCO₃), Cu-Hist at OCP, and Cu-Hist at -0.1 V with N₂ and CO₂ purging. Band broadening was observed between wet/dry histidine and Cu-Hist sample. More importantly, there are some bands that are present only with applied cathodic voltage (1626 and 2080 cm⁻¹)

As further proof that the histidine can be retained on the surface with voltage application, we performed flushing experiment where we flush electrolyte with fresh KHCO₃ whilst the applied voltage is still turned on at -0.7 V (**Figure S5.5**). The result is that, albeit with reduced intensity, we can still observe bands belonging to histidine-Cu interaction at cathodic voltage.

Figure S5.5: (a) Electrolyte purging experiment: Raman measured before and after flushing with 50 mL fresh KHCO_3 under continuous CO_2 stream and -0.7 V applied. Raman cell volume is approx. 30 mL. (b) Raman spectra before and after purging, showing persistent histidine bands after flushing.

In addition, DFT calculations demonstrate that the adsorption of histidine on (reduced) Cu substrate is energetically more favourable by -4.77 eV :

Where partially deprotonated (in the carboxylic group) histidine-Cu complex and H atom are co-adsorbed on the Cu (100) surface.

This information above is included in new SI Section 5.3.

4. Why the multiple bands (in the range of $100\text{-}1600\text{ cm}^{-1}$) attributed by the authors to histidine were not observed at low applied potential range (until -1.1 V) for the Cu-hist sample under CO_2 conditions (figure 3c). If there were a strong interaction between histidine and Cu, these bands should have been observed regardless of applied potentials.

The reviewer is correct to point out that the strong histidine-Cu interaction should be visible regardless of applied potentials. In fact, this is true. In Figure S5.3 we conducted experiment with dissolved histidine (0.025 M concentration) on bare Cu metal. Because of the high concentration, and histidine is free-floating in the electrolyte, traces of histidine-Cu interactions are already visible at open circuit potential.

However, for our *synthesised* Cu_2O -hist sample, we did not expect clear Raman signals at OCP because (1) most of the histidine should be physically bound/encapsulated over Cu_2O (see explanation on question 3), (2) there is low concentration chemically bound histidine on Cu_2O surface (as seen on XPS, SI Section 2.5), (3) Cu_2O is not an effective SERS surface. As the electrochemical voltage is applied and Cu_2O is reduced to roughened Cu, Cu-histidine bands can be seen more clearly. This is because

rough Cu surface that enhances the Raman signal are exposed, and physically bound histidine is now able to interact with reduced Cu surface.

To improve this, we have since repeated the in-situ Raman measurements and updated main text **Figure 3**, and **Figure S5.3**.

Figure 3. In-situ Raman spectroscopy on bare and histidine functionalised Cu_2O under CO_2RR relevant conditions. Comparisons were made on three different conditions to ascertain histidine presence during CO_2RR and the expected $^*\text{CO}$ binding configuration on (a) electrodeposited Cu_2O with added 0.025 M histidine dissolved in the electrolyte. (b) Cu-0 and (c) Cu-Hist. Measurements were stopped at different potentials depending on the vigorousness of the bubbling that disrupts *in-situ* Raman signal. Electrolyte: CO_2 purged 0.1 M KHCO_3 , $\text{pH} \approx 6.7$. Red shaded area: expected region of adsorbed CO_2RR intermediate bands. Blue shaded area: expected region of Cu^{1+} bands. Yellow shaded area: expected region of Cu-Histidine complex bands. Dashed lines are a guide to the eye. Raman bands marked at 1009, 1259, 1321, 1485, 1572, and 1640 cm^{-1} can be matched with Raman bands of deprotonated L-histidine adsorbed on Cu in alkaline condition under applied cathodic bias.²⁵

Figure S5.3: In-situ Raman in N₂ Purged 0.1 M KHCO₃ with dissolved 0.025 M Histidine on (a) polished bare Cu disc and (b) freshly electrodeposited Cu₂O on Cu disc. (c) *In-situ* Raman on Cu-Hist sample in N₂ purged 0.1 M KHCO₃ electrolyte. (d) *In-situ* Raman on freshly electrodeposited Cu₂O on Cu disc in CO₂ purged 0.1 M KHCO₃ electrolyte

To address questions 3 and 4, we have revised the main text in page 6 as follows:

We continued the *in-situ* Raman investigation on Cu-0 in CO₂ purged 0.1 M KHCO₃ electrolyte (**Figure 3b**). As expected, Cu-0 behaves just like typical Cu₂O-derived Cu catalysts,²⁸ where the Cu₂O bands disappear almost instantly when -0.10 V cathodic potential was applied. The expected C≡O frustrated rotation and Cu-CO bands at 279 and 364 cm⁻¹ were also observed clearly once the potential reaches -0.7 V onwards, indicating the suitability of our system to detect the signature of such intermediate species. We note that similar bands are also observed on benchmark measurement on electrodeposited Cu₂O in CO₂ purged 0.1 M KHCO₃ electrolyte (**Figure S5.3d**).

Moving on to unreduced Cu-Hist sample, only Cu₂O related bands at 519 and 620 cm⁻¹, and broad humps around 1130 cm⁻¹ were present at OCP (**Figure 3c**). The absence of histidine-related bands at

OCP is reasonable, as the initial interaction of histidine on unreduced Cu-Hist is expected to be a mixture of physical and chemical interactions (see **Figure S2.6** and **SI Section 5.3** for details). The coverage of chemically bonded histidine on unreduced Cu-Hist are relatively low, approximately 1 histidine molecule per 16-26 surface Cu atoms as inferred by XPS (**Figure S2.10**). Cu₂O related bands immediately disappeared upon -0.10 V cathodic voltage application, accompanied by appearance of CO₃²⁻ band at 1073 cm⁻¹, and a series of strong bands that closely match dissolved histidine experiment and literature values (major: 1009, 1259, 1321, 1485, 1572, 1640, 2081 cm⁻¹). These histidine related bands on Cu-Hist are markedly more intense than the ones observed on dissolved histidine experiments (**Figure 3a**), even though the effective histidine concentration in the system is much lesser.

We posit that the initial strong chemical and physical interaction between histidine and Cu₂O in the unreduced stage through Cu-N bond (inferred from additional Cu2p peak at higher binding energy and significantly shifted N1s of Cu-Hist sample, **Figure S2.9**), may be critical for achieving high surface coverage of histidine on the reduced Cu surface and enhanced CO₂RR when compared to physically mixed histidine (**Figure S4.6a**). Intriguingly, expected bands related to Cu-CO and C≡O frustrated rotation (expected around 279 and 364 cm⁻¹) are also missing in Cu-Hist, even after ramping the cathodic potential to -1.10 V. Given the excellent C₂₊ selectivity of **Cu-Hist**, persistent histidine adsorption and missing M-CO bands at very highly cathodic potentials observed through *in-situ* Raman indicate that strongly adsorbed histidine may alter the interactions between Cu surface and *CO (or related intermediates).

5. Following above question, interestingly, under N₂ conditions, above bands could be observed at much lower applied potentials (starting at -0.1 V). The authors need to explain this discrepancy.

We have repeated the *in-situ* Raman in N₂ and CO₂ purging condition, both in main text **Figure 3** and Supporting Information **Figure S5.3**, which showed much earlier Cu₂O bands disappearance, and correspondingly stronger Cu-hist peaks. Our hypothesis is that the histidine bands can only be visible once Cu₂O is reduced to Cu, enhancing the Raman signals because of roughened Cu surface.

The main change between our previous *in-situ* Raman experiment setup and the new data is that we removed the electrode top cover that reduces bubble accumulation significantly, thus allowing us to collect better data on wider surface area of the Cu-Hist samples under electrochemical bias.

In the updated data, we see no difference between onset of histidine peak appearance in CO₂ and N₂ purged condition. Note that voltage adjustment based on pH difference between N₂ and CO₂ purging has already been taken into account. Traces of histidine peaks can be observed at OCP on dissolved histidine samples, but not on Cu-Hist samples, possibly due to free-moving histidine that can approach/align to the surface better than physically bound histidine on Cu-Hist sample. This is described in **Figure R1** below compiled from **Figure 3** and **Figure S5.3** for the reviewer's convenience.

Cu₂O with 0.025 M dissolved histidine (CO₂) Cu₂O with 0.025 M dissolved histidine (N₂)

Figure R1: Comparison between CO₂ and N₂ purged Raman data on dissolved histidine and Cu-Hist samples

6. In figure 3c, the bands attributed to copper oxides persisted until potentials as negative as -0.7 V, inconsistent with the authors' claim (lines 160-161, pg5) that "Cu₂O bands disappear almost instantly when -0.10 V cathodic potential was applied". This observation has been noted by the authors (lines 177-180, pg 6) but no explanation was given. I feel this is important, as the histidine may play a role to stabilize Cu₂O – at least at low overpotential range – which is believe by some researchers to enhance C₂+ product selectivity.

We thank the reviewer to point out this. We were aware of the appearance of persistent Cu₂O bands but decided not to elaborate on it because we feel that it might be a spectator effect. After receiving feedback from Reviewer 1, we have repeated in-situ Raman three more times (see **Figure R2** below), and we observe no clear evidence of persistent Cu₂O peaks were observed. We have since updated the main text **Figure 3** and Supporting Information **Figure S5.3**, which showed much earlier Cu₂O bands disappearance, and correspondingly stronger Cu-hist peaks at earlier cathodic potential. As with questions 4 and 5, we posit that histidine bands will be visible once Cu₂O is reduced, allowing it to

break free from physical restraint and interact chemically with Cu. Roughened Cu₂O-derived Cu also enhances the Raman signals.

Figure R2: Five repeat measurements of in-situ Raman on Cu-Hist samples under different applied potentials, from OCP up to -1.1 V vs RHE.

7. Above three questions raised my concerns about quality and reliability of the obtained in-situ Raman data and associated hypothesis on *CO adsorption (e.g., lines 156-157, pg5; lines 181-184, pg6).

We thank the reviewer for raising this concern. We highlight that, despite the changes in the potentials where Cu₂O Raman bands was reduced (around 520 and 630 cm⁻¹), **none of the five repeats on Cu-Hist samples show any *CO adsorption band that is expected around 279 and 364 cm⁻¹ on Cu surface.**

We have shown on measurements on Cu-0 (Figure 3b), and baseline measurements on regular Cu₂O samples grown separately *via* electrodeposition (Fig S5.3b) that *CO can be detected on our Raman system consistently. In fact, we have shown that whenever histidine is added, either on the surface or in the electrolyte, the *CO peaks was not seen even after adding potential to -1.3 V (Figure 3a).

Thus, we are confident in our position that the presence of histidine alters intermediate adsorption on Cu surface and that *CO bands are not observed on systems containing histidine under CO₂ purged electrolyte and cathodic bias up to the voltage limitation of our Raman system at -1.1 V RHE.

8. I do not believe operando can be used to term the Raman spec study. It is at most an in-situ study – the test conditions for Raman spec were distinct from the conditions for the CO₂RR performance evaluation.

We thank the reviewer for the suggestion. We have replaced *operando* with *in-situ* Raman in all text and supporting information.

B, Other comments:

1. The Cu/N ratio and Cu-N interaction are based on characterisation of the pristine Cu₂O-histidine sample, which is different from the real catalyst (at least Cu₂O became Cu). Drawing any conclusions from such characterisation should be cautious.

We thank the reviewer for the kind reminder. Indeed, the ex-situ XPS data shown in **SI Section 2.5** shows that there is a very strong interaction between histidine and Cu₂O in the as-synthesised Cu₂O-histidine. We believe this close contact is crucial prerequisite to the strong subsequent interaction during CO₂RR as it will allow the histidine to “stick around” more and increases the surface coverage. This explains why physically mixed Cu₂O-0 and histidine at similar loading could not perform CO₂RR that well as pre-synthesised Cu-Hist (see **SI Section 4.6**).

Crucial evidence of Cu-histidine interaction under electrochemical bias is provided by the Raman. In **Fig 3 and Fig S5.3**, multiple Raman bands around 800-2200 cm⁻¹ can be tagged to Cu-N (N in the imidazole ring) interaction and histidine internal structure.

Indication of Cu-N from as-synthesised sample is encouraging to us because this indicates that the Cu-N linkage between Cu₂O and histidine is *already* occurring in the precursor stage. We expected that the Cu¹⁺ will reduce to Cu⁰ during CO₂RR (which is also seen on the revised in-situ Raman in Figure 3). However, the persistent Raman bands tagged to the histidine and Cu-Histidine interaction indicate that Cu-N interacts *during* CO₂RR relevant conditions.

2. *CO as the starting point for DFT calculations is not sound to me because: a) as this paper aimed to propose a new mechanism, it is reasonable to examine whether the histidine has any impact on elementary steps along the CO₂ to *CO coordinates, and b) no Raman signal for *CO was observed to support that *CO was even an intermediate in the Cu-Hist case.

As suggested by the reviewer, we explored a reaction pathway of the CO₂ conversion in the presence of histidine (**Figure R3**). Indeed, the CO₂ can be stabilized by histidine (**Hist**) via forming the **Hist**-CO₂ complex ($\Delta G_{1-2} = -0.68$ eV). Starting from **Hist**-CO₂ complex (**2**), sequential proton-electron transfer steps lead to **Hist**-CO (**4**) via **Hist**-COOH (**3**), where the first CO is generated with the aid of histidine. Although the adsorption of histidine lowers the chance of CO₂ adsorption on the Cu surface, it is still possible to happen during the reaction. The CO₂ decomposition on the Cu surface results in the second *CO, which prefers to adsorb beside the histidine-CO complex ($\Delta G_{4-5} = -0.16$ eV). The further protonation of *CO (**6**) and the subsequent C-C coupling between the generated *CHO and the **Hist**-CO complex produce the C₂₊ species, *i.e.*, the **Hist**-COCHO intermediate (**7**). Here, we found that the formation of *CHO is a key step ($\Delta G_{5-6} = 0.72$ eV), where an applied potential of -0.72 eV can make all electrochemical steps downhill.

Apart from the reaction mechanism, we identified three factors that may help rationalize the absence of Cu-*CO and C≡O frustrated rotation Raman bands during our experiments.

1. The CO₂ molecule preferentially binds to the histidine molecule and the derived **Hist**-CO has a decreased interaction with the surface Cu sites.
2. The estimated surface coverage of histidine (1 molecule per 16-26 surface Cu atoms) may limit the amount of surface CO bound to Cu sites, effectively decreasing the intensity of any Cu-CO Raman bands
3. At potentials more negative than -0.72 V according to our calculations, the *CHO surface species are thermodynamically more stable than *CO (CPET **4**→**5**, **Figure 4**) and their population on the surface are expected to dominate.

We have changed **Figure 4** of the manuscript (pasted below for Reviewer’s convenience). A discussion of the updated **Figure 4** has also been added on page 7 of the manuscript.

“First, a CO₂ molecule approaches the surface and physisorbs on Cu sites near the deprotonated amine group from histidine (**1**, **Figure 4**). The *CO₂ adsorbate may bind the N atom in histidine (**1**→**2**) by overcoming a barrier of 0.23 eV to form the **Hist**-CO₂ complex (**2**). The C-N coupling is highly

exergonic with $\Delta G_{1\rightarrow 2}$ of -0.68 eV, indicating a thermodynamically favoured product. The electrochemical conversion of Hist-CO₂ to Hist-CO involves two coupled proton-electron transfer (CPET) steps. The first CPET forms **Hist-COOH (3)** in a slightly endergonic process ($\Delta G_{2\rightarrow 3} = 0.05$ eV), while a subsequent CPET generates H₂O and **Hist-CO intermediate (4)** on the surface ($\Delta G_{3\rightarrow 4} = 0.66$ eV). From here, a surface *CO, originally on Cu sites distant from the histidine molecule, approaches to the **Hist-CO intermediate (5)**. The free energy change ($\Delta G_{4\rightarrow 5} = -0.16$ eV) suggests that the approach of *CO to sites near **Hist-CO** may occur spontaneously. A following CPET would transform the *CO into *CHO (**6**) with $\Delta G_{5\rightarrow 6}$ of 0.72 eV, which is the most endergonic step, thus a modest potential should be applied to make the reaction proceed (**Figure 4a**, red curve). Once the generated *CHO (thermodynamically favoured over *COH on Cu(100)³¹) is present around the **Hist-CO** intermediate, the C-C coupling between the histidine-bound *CO and *CHO (**6**→**7**, **Figure 4b**) is both kinetically ($E_a = 0.33$ eV) and thermodynamically ($\Delta G_{3\rightarrow 4} = -0.78$ eV) more favourable than the baseline cases (*CO-*CO and *CO-*CHO coupling on Cu(100) surface, **SI section 6.8**)."

In addition, possible factors affecting the Cu-*CO Raman bands have been added on page 8 of the manuscript.

"This novel mechanism for CO₂RR via histidine-assisted transformations may help rationalizing the absence of the C=O frustrated rotation in the Raman bands at applied bias during our experiments. On the one hand, *CO₂ may be transformed into *CO while bound to the histidine molecule through the amine N atom (**1**→**2**→**3**→**4**), limiting the interaction of *CO with the Cu surface sites. On the other hand, the estimated high surface coverage of histidine (1 molecule per 16-26 surface Cu) may limit the amount of surface *CO bound to Cu sites, considerably affecting the intensity of characteristic Raman bands. In addition, the resulting few Cu-*CO intermediates may transform quickly into *CHO at high negative potentials (<-0.72 V) according to the Boltzmann probability distributions (**SI section 6.5** and **Figure S6.3**), reducing further the measurable Cu-*CO indicators."

Figure 4: Initial reaction steps during CO₂RR over histidine-Cu/Cu(100) substrate calculated by DFT. (a) Gibbs free energy (GFE) diagram and the (b) snapshots of the first few surface intermediates in the histidine-assisted CO₂RR mechanism. The GFE diagram was calculated from the reference state (0), histidine-Cu/Cu(100) shown in Figure S6.2b, a gas-phase CO₂ molecule and an adsorbed *CO. In configurations (1-4), the *CO molecule adsorbed on a bare Cu(100) substrate is omitted for clarity, however, the energy has been added to each system accordingly. This *CO approaches the Hist-CO intermediate (5) and becomes *CHO after a CPET step (5→6), where the thermodynamic barrier $\Delta G_{5\rightarrow6}$ of 0.72 eV can be overcome with applied bias (red line). The newly formed *CHO species may couple with the co-adsorbed Hist-CO intermediate through a C-C bond to form 7. The intermediates of surface reactions (1→2 and 6→7) are connected with smooth lines from which the energy level of the TS may be inferred (the highest point of the smooth lines).

3. Water solvent should be used and possible hydrogen-bond interactions should be considered in the DFT calculations, in particular, in the presence of additional molecule layer.

We thank the reviewer for pointing this out. Based on the suggestion, we investigated and compared various solvation methods and their effects on the stabilization of critical intermediates (SI Section S6.6). The Figure S6.5 presents the results of such comparison. In short, we considered that the widely used semi-empirical approach (presented as SE1, reference 22-26 in the supporting information) can adequately describe the solvation energy of the relevant intermediates, thus it was adopted in our work. This semi-empirical approach is based on calculations with explicit water molecules, deriving the solvation correction for *OH, *R-OH, and *CO (carbonyl-containing) surface species, thus H-bond interaction has been considered implicitly in this approach.

We added Figure S6.5 and corresponding discussions in SI Section S6.6. The energies of all intermediates in Figure 4 and Figure 5 have been corrected by using semi-empirical approach (SE1).

4. It was not explained by DFT calculations why ethanol selectivity was largely promoted in the experiment (lines 116-118, pg4, and figure 2).

As shown in Figure 2a and 2b of the main text, we can see that the production of C₂₊ products (particularly ethanol and ethylene) on Cu-Hist rises significantly with increasing voltage compared to the Cu without functionalisation groups, i.e., Cu-0.

In order to understand the effect of the histidine, we designed the DFT work to investigate the reaction mechanism comparing CO₂RR towards CH₄ (representing C₁ compounds) and C₂₊ compounds. As the amount of ethylene and ethanol produced on the Cu-Hist catalyst is about the same at all working potentials (Figure 2b), in our simulation work, we took C₂H₄ as an example to represent the C₂₊ products and focused on comparing the selectivity of CH₄ and C₂H₄ over Cu-Hist and Cu-0 catalysts.

We agree with the reviewer that a higher ethanol yield can be observed on Cu-Im and Cu-ImPA (Figure 2c-d). However, due to the lower peak FE (Figure 2c-d) and current density (*j*) (Figure S4.5b) than Cu-Hist, Cu-Im or Cu-ImPA is not our research focus. Thus, the investigation on possible bifurcations favouring ethanol over Cu-Im and Cu-ImPa is outside the scope of our computational work, although this topic is interesting and worthwhile to be explored in the future.

To clarify this point, we have revised on page 7:

“We turn to DFT calculations to rationalize the effects of histidine on the selectivity towards CH₄ and C₂H₄ over the Cu-Hist catalyst”

Reviewer #2 (Remarks to the Author):

This manuscript studied organic-functionalized Cu with improved C₂⁺ products in electrochemical CO₂ reduction. The organics include histidine, imidazole, 2-78 methylimidazole, imidazolepropionic acid, arginine, triazole and glycine. Some characterizations were carried out, such as Operando Raman, DFT calculations, EIS, mPV, to explore the mechanism on the interface or catalyst/electrode surface. The decoration method is interesting, which realizes high and stable C₂⁺ product selectivity.

We thank Reviewer 2 for the time taken to evaluate our manuscript carefully.

However, some discussions and conclusions need careful deduction. The surface charge was considered to be universal activity descriptor to explain CO₂RR activity and proxy to catalyst development. The integrated anodic transient charge (Q_{an}), derived initial surface charge accumulation (C_{2nd}) and desorption rate (k_{2nd}), are all correlated to C₂⁺ selectivity. These parameters seem to obtain from kinetic functions (kinetic parameter). It is rough and limited to conclude. What kind of charge on surface, from what surface species? The organics functionalized Cu and pristine Cu (Cu₀) are comparable to use surface charge descriptor.

We speculate that the surface charge measured here may be related to local electrostatic interactions within double layer and interface. Such interactions have been proposed to be the possible reason of shifting CO₂RR preference in presence different alkali cation in the electrolyte (e.g., Monteiro, M. C. O. *et al. Nat. Catal.* **4**, 654-662, doi:10.1038/s41929-021-00655-5 (2021).; and also Waegele, M. M., Gunathunge, C. M., Li, J. & Li, X., *J. Chem. Phys.* **151**, 160902, doi:10.1063/1.5124878 (2019).)

Unlike cations, whose interaction has been shown to be confined to non-covalent (e.g., Strmcnik, D. *et al., Nat. Chem.* **1**, 466, doi:10.1038/nchem.330 (2009)), histidine is rather unique because it is known to form specific adsorption on Cu through either carboxylic or imidazole nitrogen. (shown through our in-situ Raman data)

However, we acknowledge that, like most other electrochemical techniques, our proposed mPV measurements are not able to identify the adsorbed species, whether they have specific adsorption to the surface or the adsorption behaviour. We developed it as a simple way to estimate the accumulation of all charged species in the double layer and on the catalyst surface that is related to the CO₂RR activity. Further work on extending the technique to other samples and conditions are under way.

For this we changed the text in page 12 after **Figure 6**:

In search for a more ideal way to measure surface charge, we turn to modified pulsed voltammetry (mPV) technique (see **SI Section 8** for details). The mPV excitation pulse was constructed with a fixed upper bound near the OCP (typically around +0.2 to +0.3 V) and gradually more cathodic lower bound voltage between around -0.5 to -1.6 V which is relevant to the CO₂RR operating voltage of our catalysts. The applied mPV would trigger a repeating cycle of charged species accumulation (adsorption) at the cathodic lower bound and decumulation (desorption) at the upper bound near OCP. Like EIS and other electrochemical techniques, we recognise that the mPV cannot identify adsorbed intermediate species, whether they are adsorbed specifically, or the adsorption behaviour.^{37,41,42} However, it is able to give a support to their existence and give some estimate of nett charge and kinetic parameters of the desorption of such species when the applied cathodic potential is removed.

On the other hand, the Operando Raman, DFT calculations, EIS, mPV were carried out separately, but a systematical discussion is highly encouraged, even partially. Are there any relationships among results from different characterization?

Thank you for your input. We have now expanded the discussion to link on the in-situ Raman, DFT and transient electrochemical measurements together. For example, DFT calculations were constructed based on the operando Raman data showing that the adsorbed histidine is persistent throughout CO₂RR condition. The dynamic electrochemical techniques were then performed to measure surface charge changes, such adsorbed species may alter local electric field around the electrode, as observed in the cation effect.

We have now added new systematic discussion to link the physical and electrochemical characterisation with theoretical calculations.

Page 10 after Figure 5

From *in-situ* Raman and DFT investigations we learnt that histidine remains specifically adsorbed on Cu-Hist surface during CO₂RR and how it provides alternative pathways towards C₂₊ via formation of Hist-CO. The presence of such surface adsorbed species may change the local electric field and provide new electrostatic interaction near the catalyst surface, akin to the cation effect reported previously.³⁴⁻³⁶ Therefore, we enlist transient electrochemical techniques in attempt to quantify the changes in the surface charge. At the same time, we posit that the metrics obtained by transient electrochemical techniques may also be exploited as “activity descriptors” to predict CO₂RR activity on wide range of catalyst systems.

Page 12 after Figure 6

In search for a more ideal way to measure surface charge, we turn to modified pulsed voltammetry (mPV) technique (see **SI Section 8** for details). The mPV excitation pulse was constructed with a fixed upper bound near the OCP (typically around +0.2 to +0.3 V) and gradually more cathodic lower bound voltage between around -0.5 to -1.6 V which is relevant to the CO₂RR operating voltage of our catalysts. The applied mPV would trigger a repeating cycle of charged species accumulation (adsorption) during cathodic lower bound voltage and decumulation (desorption) at OCP. Like EIS and other electrochemical techniques,^{36,40-42} we recognise that the mPV by itself cannot positively identify adsorbed intermediate species and their adsorption behaviour. However, it is able to give a support to their existence and give some estimate of the nett charge and kinetic parameters of the desorption of such species when the applied cathodic potential is removed.

Page 17 after Figure 7

Parallels can be drawn between our results that shows clear CO₂RR activity shift in presence of histidine to the cation effect on CO₂RR selectivity.³⁴⁻³⁶ A major difference between the cation effect and our Cu-Hist is that the histidine can be specifically adsorbed to Cu₂O-derived Cu surface under cathodic bias, as shown by the *in-situ* Raman data (**Figure 3** and **S5.3**). Further, the hydration shell of histidine (and many other amino acids) are much larger than alkali cations, but softer,⁴⁴ thus enabling higher surface charge accumulation than Cs⁺ cation while allowing for specific adsorption to reduced Cu surface. In addition, much stronger interaction with intermediate is afforded on Cu-Hist through amine group, unlike cations where only electrostatic effects are available.³⁶ To increase the surface charge, we speculate that the amino acids could be modified by placing electron donating groups on the amino nitrogen to increase basicity, whilst balancing the interaction with *CO or *CHO intermediate coupling. One could also combine the effect of cation together by swapping K⁺ with Cs⁺ that may allow further stabilisation of intermediate through electrostatic effect.

To construct a universal descriptor is important but difficult. The scope of application is especially important. I doubt the study and conclusion apply only to the catalyst system in this manuscript.

We agree with the reviewers' assessment. The current work confirms surface charge increase on organic functionalised Cu surface, and we have demonstrated that the correlation to CO₂RR activity extends to wide array of organic functional groups. As described in new text in Page 17, further studies on the application of such activity descriptor to other catalyst surface is under way.

Reviewer #3 (Remarks to the Author):

The manuscript by Handoko et al reports on the mechanistic understanding of Cu/organic electrocatalysts towards the reduction of CO₂ to C₂+ products. Under normal circumstances I would consider this an extremely well-trodden territory as the electroreduction of CO₂ by Cu was first reported by Hori about 30 yrs ago and has become a very hot topic in the last decade. Such a conclusion would be premature as there are many aspects of this paper which bring a fresh take to an old story. The strengths of this paper are:

A, The careful exploration of multiple catalyst systems to determine one which is both selective towards c₂ species and stable (for at least 48hrs).

B, Unique analysis on the catalytic performance using time resolved electrochemical measurements to extract descriptors of reactivity.

C, The surface charge descriptor is in good accord with chemical intuition toward activity and could serve as a useful tool for future design principles.

I believe this paper could make a solid addition to nature comm. but a few things need to be strengthened first.

I would suggest the authors consider the following issues to improve the overall impact of their work:

A, Can the authors discuss the activity of their catalyst in terms of turn over frequencies (TOF) to better gauge them against thermal catalysts and project on the potential to operate at high current densities (ca A/cm²) which would be the requisite for utilization of this catalysts for practical deployment.

Admittedly, the TOF demonstrated in our work is small, around 2.9E-3 conservative estimate for total C₂+ product per total number of Cu atoms (assuming all Cu atoms are active site) or to 4.2E-1 (assuming all histidine sites are active) at -2 V vs RHE. This is because the work is aimed at fundamental understanding and the cell used is not designed to handle high current densities.

Projected to higher current density of 1 A/cm² achievable on GDE type electrodes, the TOF is estimated to be between 0.19 to 27.2 of C₂+ products. Works on demonstrating higher current density is in progress.

TOF calculation is added in new Supporting Information Section 3.3

B, The theoretical approach uses standard approaches based on the computational hydrogen electrode as popularized by Norskov and co-workers. None the less the results could be more effectively presented as a catalytic cycle where the energetics (at zero and higher applied bias is included in for each step. This is particularly true to the data in figure 5 which is less accessible. Can the authors comment on the expected surface abundance of CO and Formyl groups. The latter of which their energetics indicates is in low abundance yet is being invoked in a surface reaction with X bound CO. Can the authors comment on the potential role of solvent and electrolyte which are not included but are known to change energetics (sometimes substantially).

We thank the reviewer for the constructive suggestions for further improving our study.

1. We have now added **Figure S6.3** which presents the effect of applied potential in the CO₂RR via the histidine-assisted mechanism (**Figure 5**). The applied potential changes the relative energy level of C-N coupling (**1** to **2**) and C-C coupling (**6** to **7**) on the energy profile, leading to a change of the key step during the chemical process.

We added the corresponding description on page 8 of the manuscript.

“We found that the Hist-CO and *CHO coupling (6 to 7) has the highest energy level on the energy profile, which is the key step at 0 V (Figure 5a). However, the CO₂ binding with Cu-Hist (1 to 2) becomes more important when a potential of -0.72 V is applied (Figure S6.3).”

2. With respect to the question on the abundance of surface *CO and formyl groups (*CHO), this is related to Reviewer-1's question B-2. Although the *CHO species are thermodynamically less stable than *CO on Cu(100), their abundance could increase at high negative potentials. We have calculated the Boltzmann probability distributions of *CO and *CHO at various applied potentials, the ratio of probabilities depends only on their energy difference (Figure S6.3). Our calculations indicate that *CHO population should surpass that of *CO at potentials more negative than -0.72 V, considering no kinetic or diffusion limitations.

We have added the following discussion on page 9 of the manuscript, and SI Section 6.5 with more details on this topic:

“In addition, the resulting few Cu-*CO intermediates may transform quickly into *CHO at high negative potentials (<-0.72 V) according to the Boltzmann probability distributions (SI section 6.5 and Figure S6.3), reducing further the measurable Cu-*CO indicators.”

3. With respect to the potential role of the solvent, we explored how typical intermediates are stabilized by various solvation correction schemes in our computational study, including two semi-empirical approaches (SE1 and SE2) based on explicit solvation models and one implicit solvation model (VASPsol). We found that the intermediates are stabilized by -0.19 eV (SE1), -0.17 eV (SE2), and -0.22 eV (IS) in average with respect to the calculations in vacuum with a consistent standard deviation of 0.09 eV. The three approaches show the same varying trends and similar results for the solvent effect. Therefore, we used the SE1 approach for all calculated intermediates in this work, which is good enough to describe the solvation energy.

We added Figure S6.5 and corresponding discussions in SI Section S6.6. The energies of all intermediates in Figure 4 and Figure 5 have been corrected by using semi-empirical approach (SE1)

In addition, the role of the electrolyte might be important, but the methods used to describe it are not yet mature.[see e.g., Abidi, N., Lim, K. R. G., Seh, Z. W. & Steinmann, S. N., *WIREs Computational Molecular Science* **11**, doi:10.1002/wcms.1499 (2021).]

Because configuration sampling becomes a challenge with the addition of weakly bound species to the electrode (i.e., solvent, electrolyte), accurate treatment of the solid-liquid interface often requires obtaining statistical averages over expensive *ab initio* molecular dynamics (AIMD) simulations.[Sundararaman, R., Vigil-Fowler, D. & Schwarz, K., *Chem. Rev.* **122**, 10651-10674, doi:10.1021/acs.chemrev.1c00800 (2022).] Continuum solvation methods can approximate these statistically averaged solvation effects, but do not account for the electronic interactions between the electrode and electrolyte. Thus, the electrolyte effect was not considered in this simulation work.

C, In the conclusion section the authors need to stretch themselves and project on how their finding can be used to enhance reactivity. Can they discuss methods for boosting surface charge, ex increasing the basicity of the amine, changing the ion content in the solvent layer adjacent to the surface etc. I feel this is a sadly missed opportunity to impact an ongoing conversation that has been going for a long time.

Thank you for the suggestion. We agree that it is important to suggest methods to modulate the surface charge further. We see some parallel between previously reported cation effect to our work, but in this case, we propose that histidine can bring higher impact to the CO₂RR yield because of its ability to be specifically adsorbed, with larger hydration shell but softer compared to large cations.

We added a discussion in page 14 after Figure 7:

Parallels can be drawn between our results that shows clear CO₂RR activity shift in presence of histidine to the cation effect on CO₂RR selectivity.^{40,41} Murata and Hori realised the cationic effect early,⁴² noticing the difference in cation effect to CO₂RR performance in the order of (Li<Na<K<Rb<Cs). It is still unclear if cations are specifically adsorbed to catalyst surface under cathodic bias,^{43,44} and has proposed to alter the CO₂RR via modification of the local electric field, buffering of the interfacial pH or stabilization of reaction intermediates.⁴¹

A major difference between cations and histidine, is that the histidine can be specifically adsorbed to the catalyst surface under cathodic bias, as shown in in-situ Raman data. The hydration shell of histidine (other amino acids) are much larger but softer,⁴⁵ thus enabling higher surface charge accumulation than Cs⁺ cation, but still allowing for specific adsorption to reduced Cu surface. In addition, appropriate ligand connection through the amine group allows for stronger interaction with intermediate, unlike electrostatic effect available in the case of cations.⁴⁶ To increase the surface charge, the amino acids could be modified by placing electron donating groups on the amino nitrogen to increase basicity, while balancing the interaction with *CO or *CHO intermediate coupling. One could also combine the effect of cation together by swapping K with Cs cation that allows further stabilisation of intermediate through electrostatic effect.

REVIEWERS' COMMENTS

Reviewer #1 (Remarks to the Author):

The authors have done a great work in addressing most of my comments. The only remaining area I see further improvement before a recommendation to publish can be made is theoretical study. The introduction of histidine molecule makes the calculation of reaction pathway complex. As histidine plays important role in altering product selectivity, I feel it would be a good addition if the authors could include more detailed discussion on possible alternative reaction pathways and in particular on why the proposed reaction path in Figure 4 is the most energetically favorable.

Reviewer #2 (Remarks to the Author):

The authors have addressed all my concerns. I recommend the manuscript's publication.

Reviewer #3 (Remarks to the Author):

the revisions are satisfactory

Reviewer #1:

Remarks to the Author:

The authors have done a great work in addressing most of my comments. The only remaining area I see further improvement before a recommendation to publish can be made is theoretical study. The introduction of histidine molecule makes the calculation of reaction pathway complex. As histidine plays important role in altering product selectivity, I feel it would be a good addition if the authors could include more detailed discussion on possible alternative reaction pathways and in particular on why the proposed reaction path in Figure 4 is the most energetically favorable.

We thank the reviewer for their positive assessment and final suggestion. Accordingly, we have added three sentences to the results section to reflect other intermediates studied and the selection of the most likely pathway based on thermodynamic barriers (highlighted in yellow, page 8 of the main document). In addition, a diagram showing alternative reaction channels was added in SI Section 6.9

FROM THE RESULTS SECTION OF MAIN TEXT:

We then turn to DFT calculations to rationalise the effects of histidine on the selectivity towards CH₄ and C₂H₄ over the Cu-Hist catalyst. A single deprotonated histidine molecule was placed over a 4×4 Cu(100) surface slab, based on the observed state and coverage of histidine in our synthesized catalyst from the XPS (SI section 2.5) and Raman data (Figure 3c and SI Section 5). Details of the surface model optimisation are described in SI Section 6.2. It is widely accepted that the CO₂RR on Cu proceeds through a common *CO intermediate.³¹ However, our experimental results indicate a possible deviation from the typical *CO–catalyst interaction (absence of Cu–*CO and C≡O frustrated rotation). Accordingly, we explored an alternative *CO₂ to *CO conversion with subsequent transformation into CH₄ and C₂H₄ with direct involvement of a histidine molecule. In the following discussion, all intermediates during the reactions are labelled with bold numerals, while each elementary step is represented by A→B, where (A) and (B) are two consecutive intermediates. In the description of reaction intermediates, the “Hist” label refers to the co-adsorbed histidine molecule. We discuss the thermodynamics of the transformation in terms of Gibbs free energies (G), and Gibbs free energy change ($\Delta G_{A\rightarrow B}$).

First, a CO₂ molecule approaches the surface and physisorbs on Cu sites near the deprotonated amine group from histidine (1, Figure 4). The *CO₂ adsorbate may bind the N atom in histidine (1→2) by overcoming a barrier of 0.23 eV to form the Hist–CO₂ complex (2). The C–N coupling is highly exergonic with $\Delta G_{1\rightarrow 2}$ of -0.68 eV, indicating a thermodynamically favoured product. The electrochemical conversion of Hist–CO₂ to Hist–CO involves two coupled proton-electron transfer (CPET) steps. The first CPET forms Hist–COOH (3) in a slightly endergonic process ($\Delta G_{2\rightarrow 3} = 0.05$ eV), while a subsequent CPET generates H₂O and Hist–CO intermediate (4) on the surface ($\Delta G_{3\rightarrow 4} = 0.66$ eV). From here, a surface *CO, originally on Cu sites distant from the histidine molecule, approaches to the Hist–CO intermediate (5). The free energy change ($\Delta G_{4\rightarrow 5} = -0.16$ eV) suggests that the approach of *CO to sites near Hist–CO may occur spontaneously.

A following CPET would transform the *CO into *CHO (6) with $\Delta G_{5\rightarrow 6}$ of 0.72 eV, which is the most endergonic step where a modest applied potential should be applied to make the reaction proceed (Figure 4a, red curve). Once the generated *CHO (thermodynamically favoured over *COH on Cu(100)³²) is present around the Hist–CO intermediate, the C–C coupling between the histidine-bound *CO and *CHO (6→7, Figure 4b) is both kinetically ($E_a = 0.33$ eV) and thermodynamically

($\Delta G_{3\rightarrow4} = -0.78$ eV) more favourable than the baseline cases (*CO-*CO and *CO-*CHO coupling on Cu(100) surface, SI section 6.8). In our calculations, the *CO-*CO coupling over Cu(100) is endergonic by 0.96 eV and has a free energy barrier of 1.31 eV (Figure S6.8), while the coupling between *CO and *CHO is endergonic by 0.10 eV and has a barrier of 0.63 eV.

Apart from the *CO and *CHO adsorbed on the surface, other C-C coupling alternatives from (5) were also explored (e.g., coupling between (5) and *CO or (6) and *CO). However, the resulting activation energies were found to be considerably higher (1.13 eV and 1.43 eV, SI Section S6.9) than the (6→7) coupling. In addition, once the Hist-CO-CHO (7) is formed, it may be transformed into Hist-COH-CHO, Hist-CHO-CHO, Hist-CO-CHOH, or Hist-CO-CH₂O during the following CPET (SI Section S6.9). We found that the reaction is most likely to proceed via Hist-CO-CH₂O (8, Figure 5a) due to comparatively higher thermodynamic barriers in other reaction channels.

In Figure 5a, we can see all CPET steps following (7) are not potential limiting, as most of them are downhill steps and the ΔG for uphill steps (11→12) and (13→14) are less than 0.72 eV ($\Delta G_{5\rightarrow6}$). After the Hist-CH₂CH₂ intermediate (14) is formed, it needs to be decoupled from the histidine fragment to complete the whole reaction. This step requires overcoming an energy barrier of 0.79 eV to break the C-N bond (14→15). However, this C-N bond cleavage (14→15) has a much lower energy level compared to the C-C bond coupling (6→7) (Figure 5a), thus it should not be rate limiting. We found that the Hist-CO and *CHO coupling (6→7) has the highest energy level on the energy profile, which is the key step at 0 V (Figure 5a). However, the CO₂ binding with Cu-Hist (1→2) becomes more important when a potential of -0.72 V is applied (Figure S6.3).

As CH₄ formation is suppressed on Cu-Hist, a reaction pathway producing methane was also studied from (4) (Figure 5b, grey shaded substrate and SI section S6.7) to understand the reason behind this. We found this pathway requires a surface reaction (C-N bond-breaking, 18→19) to release CH₂O from adsorbed histidine for the subsequent C protonation to produce CH₄. This step has an energy barrier of 0.99 eV, making it the rate-limiting step in the formation of CH₄ on Cu-Hist, and higher than the barrier of 0.57 eV for the rate-limiting (*CO→*CHO) on bare Cu (100) (Figure S6.6).

Ergo, the presence of histidine plays opposite roles for C₂H₄ and CH₄ formation. In addition, the limited availability of H⁺ near the surface (due to the expected basic pH in the double layer) and the increased concentration of *CO species near histidine may also account for the more favourable C₂ product pathway.

This alternative mechanism for CO₂RR via histidine-assisted transformations may help rationalise the absence of the C≡O frustrated rotation in the Raman bands at applied bias during our experiments. On the one hand, *CO₂ may be transformed into *CO while bound to the histidine molecule through the amine N atom (1→2→3→4), limiting the interaction of *CO with the Cu surface sites. On the other hand, the estimated high surface coverage of histidine (1 molecule per 16-26 surface Cu) may limit the amount of surface *CO bound to Cu sites, considerably affecting the intensity of characteristic Raman bands. In addition, the resulting few Cu-*CO intermediates may transform quickly into *CHO at high negative potentials (<-0.72 V) according to the Boltzmann probability distributions (SI section 6.5 and Figure S6.4), reducing further the measurable Cu-*CO indicators.

FOR THE SUPPLEMENTARY INFORMATION FILE

6.9 Alternative pathways during the histidine-assisted hydrogenation of CO₂

We studied alternative pathways after the formation of the Hist-CO intermediate, X-CO intermediate in **Figure SError! No text of specified style in document..1**. We favoured pathways with exothermic reactions. On the most favourable pathway forming C₂H₄ (orange substrate, **Figure SError! No text of specified style in document..1**), the only endothermic reactions are the C—C coupling reaction ($\Delta E = 0.38$ eV) and the desorption of C₂H₄ ($\Delta E = 0.77$ eV). Other pathways lead to less stable intermediates. Similarly, we sampled the stability of several alternatives during the formation of CH₄ (grey substrate, **Figure SError! No text of specified style in document..1**). We identified that the most favourable pathway proceeds via the X-CO → X-CHO → X-CH₂O sequence, followed by the detachment of *CH₂O from the histidine complex.

Figure SError! No text of specified style in document..1. Studied intermediates for the transformation of the Hist-CO complex (a blue X denotes the histidine fragment) to either CH₄ (dark grey) or C₂H₄ (orange). The intermediates connected via reactions involved in the two main pathways are indicated with solid arrows, color-coded as blue for surface reactions and green for coupled proton-electron transfer reactions. Alternative reactions are also displayed with dotted arrows and their corresponding configurations. The reaction energies are indicated in black (exothermic) and red (endothermic) numbers, while the activation energies for surface reactions are shown in blue numbers below the reaction energies. Desorbing species during a reaction are indicated with labels in red font.